# All-optical control of high-purity trions in nanoscale waveguide

Hyeongwoo Lee [1], Yeonjeong Koo[1], Shailabh Kumar[2,3], Yunjo Jeong[4], Dong Gwon Heo[5], Soo Ho Choi [6], Huitae Joo[1], Mingu Kang [1], Radwanul Hasan Siddique[2,3], Ki Kang Kim[6,7], Hong Seok Lee [5] ✉, Sangmin An[5], Hyuck Choo [2,8] ✉ & Kyoung-Duck Park [1] ✉

The generation of high-purity localized trions, dynamic exciton–trion inter-conversion, and their spatial modulation in two-dimensional (2D) semiconductors are building blocks for the realization of trion-based optoelectronic devices. Here, we present a method for the all-optical control of the exciton-to-trion conversion process and its spatial distributions in a $MoS_2$ monolayer. We induce a nanoscale strain gradient in a 2D crystal transferred on a lateral metal–insulator–metal (MIM) waveguide and exploit propagating surface plasmon polaritons (SPPs) to localize hot electrons. These significantly increase the electrons and efficiently funnel excitons in the lateral MIM waveguide, facilitating complete exciton-to-trion conversion even at ambient conditions. Additionally, we modulate the SPP mode using adaptive wavefront shaping, enabling all-optical control of the exciton-to-trion conversion rate and trion distribution in a reversible manner. Our work provides a platform for harnessing excitonic quasiparticles efficiently in the form of trions at ambient conditions, enabling high-efficiency photoconversion.

The spatial control of excitonic quasiparticles in two-dimensional (2D) semiconductors has been extensively studied for the development of various exciton-based optoelectronic devices, especially facilitating intermedium of the electronic system and optical system, as well as highly efficient light-harvesting devices[1-4]. The generation of drift-induced exciton flux using various strain gradient geometries has been widely adopted in manipulating the spatial distributions of excitonic quasiparticles in transition metal dichalcogenide (TMD) monolayers (MLs)[5-7]. However, because thermally driven exciton diffusion significantly reduces exciton flux, the funneling efficiency of a neutral exciton ($X_0$) at room temperature can be very low[5]—as low as < 3%, according to a recent experimental study[6]. Meanwhile, with n-type

TMD MLs under a similar strain gradient geometry, the excess electrons are funneled together with $X_0$ and converted to trions (X-) via an exciton-to-trion conversion process. The efficiency of this exciton-to-trion conversion can reach 100% in a $WS_2$ ML suspended on a microhole-based strain gradient, because strain-induced modification of the bandgap increases the spatial overlap between $X_0$ and electrons[6] and affects Fermi level in the way of decreasing Schottky barrier height[8]. Given the characteristics of X-, particularly the high-efficiency generation and reactivity to the external electric field, the exciton-to-trion conversion can be a promising alternative to the inefficient funneling process of $X_0$. In comparison with the large-area electrical[9] and chemical doping[10] methods, exploiting strain gradient geometry

[1]Department of Physics, Pohang University of Science and Technology (POSTECH), Pohang 37673, Republic of Korea. [2]Department of Medical Engineering, California Institute of Technology (Caltech), Pasadena, CA 91125, USA. [3]Meta Vision Lab, Samsung Advanced Institute of Technology (SAIT), Pasadena, CA 91101, USA. [4]Institute of Advanced Composite Materials, Korea Institute of Science and Technology, Jeonbuk 55324, Republic of Korea. [5]Department of Physics, Research Institute of Physics and Chemistry, Jeonbuk National University, Jeonju 54896, Republic of Korea. [6]Center for Integrated Nanostructure Physics, Institute for Basic Science (IBS), Suwon 16419, Republic of Korea. [7]Department of Energy Science, Sungkyunkwan University (SKKU), Suwon 16419, Republic of Korea. [8]Advanced Sensor Lab, Device Research Center, Samsung Advanced Institute of Technology (SAIT), Suwon 16678, Republic of Korea. ✉e-mail: hslee1@jbnu.ac.kr; hyuck.choo@samsung.com; parklab@postech.ac.kr

facilitates higher conversion efficiency, local injection of the electrons, and efficient spatial controllability, suitable to applications in nanoscale optoelectronic devices and trionic energy harvesting.

However, at ambient conditions, $H_2O$ and $O_2$ molecules physisorbed onto the TMD ML surface significantly reduces electron density[11,12]. Consequently, previous studies reported only minor portions of X- in their radiative emissions compared to the dominant $X_0$ contributions, despite a 100% exciton-to-trion conversion efficiency, i.e., $X_0$ cannot be completely converted to X- at ambient conditions because of a lack of electrons[6,13]. Moreover, exploiting mechanical deformations to induce the proposed microscale 0D strain gradient can be invasive in 2D crystals with limited durability. Moreover, it restricts the direction of exciton flux and has a size mismatch with nanoscale electronics in integrated circuits. Therefore, a noninvasive and direction-controllable nanoscale platform with robust trion generation at ambient conditions is desirable for the practical application of trionic devices.

Here, we present a versatile method for the all-optical control of trion behavior in $MoS_2$ ML, including complete exciton-to-trion conversion and localization, dynamic exciton–trion interconversion, and spatial modulation of trions at ambient conditions. In our device, the nanogap geometry of the lateral plasmonic metal–insulator–metal (MIM) waveguide induces a 1D nanoscale strain gradient in the suspended $MoS_2$ ML. The induced nanoscale strain gradient significantly increases the funneling efficiency, thus confining $X_0$ to the nanogap center[14,15]; however, as described earlier, there is a lack of electrons. Therefore, the surface plasmon polariton (SPP) mode of the plasmonic lateral MIM waveguide is utilized. The plasmon-induced hot electron generation process enables the injection of electrons from Au to the $MoS_2$ ML[16,17]. These extra electrons are funneled toward the nanogap center and locally increase the electron density, stimulating additional exciton-to-trion conversion in the nanoscale region, i.e., the nanoscale generation of radiative X- emission. Thus, we can either induce complete conversion from $X_0$ to X- by activating the SPP mode or enable dominant $X_0$ by deactivating the SPP mode, i.e., achieving polarization-controllable exciton–trion interconversion between the dominant $X_0$ emission and high-purity X- emission.

Furthermore, we employ adaptive wavefront shaping using a spatial light modulator (SLM) to dynamically manipulate the SPP mode of the waveguide[18,19], which in turn spatially modulates the exciton-to-

trion conversion region[20]. We use a stepwise sequence feedback algorithm to enhance the plasmon intensity up to ~210% in the weak SPP region and observe the corresponding dramatic increase in X-emission intensity. Finally, we include the SPP effect in the drift-diffusion model to investigate the quantitative localized electron density under a nonhomogeneous strain profile. By including the X-/$X_0$ ratio from experimental photoluminescence (PL) spectra and the mass action model, we estimate an enhancement of ~10 times the localized electron density for the full activation of the SPP mode.

## Pre-characterization of all-optical trion control platform

To achieve a complete exciton-to-trion conversion and all-optically modulate their spatial distribution, we use a nanogap-based lateral MIM waveguide device with adaptive excitation control, as shown in Fig. 1a. When the naturally n-doped $MoS_2$ ML is transferred to the nanogap of the waveguide, the generated nanoscale strain gradient funnels $X_0$ together with electrons, leading to the formation of X- at the center of the nanogap (Supplementary Fig. 1). However, at ambient conditions, the electron density of the $MoS_2$ ML noticeably decreases owing to the presence of $H_2O$ and $O_2$ molecules physisorbed onto the $MoS_2$ ML surface (Supplementary Fig. 2)[11,12]. Consequently, the number of electrons is much smaller than the number of funneled $X_0$ at the nanogap, resulting in incomplete exciton-to-trion conversion. By contrast, the proposed nanogap-based lateral MIM waveguide device with the designed SPP mode can supply extra electrons locally via plasmon-induced hot electron generation, as illustrated in Fig. 1b (Supplementary Fig. 3)[16,17]. Hot electrons injected from the Au to the $MoS_2$ ML in the SPP mode drift toward the nanogap center together with $X_0$, as shown in Fig. 1c, d (a detailed description of the physical mechanism is presented using the theoretical model and experimental data in Fig. 5). This additional provision of electrons in the SPP mode results in a highly enhanced X- density and even leads to complete exciton-to-trion conversion. The polarization-sensitive nature of the SPP mode offers precise control of the exciton-to-trion conversion ratio as a function of the SPP strength, and thus dynamic interconversion between high-purity X- emission and dominant $X_0$ emission. Moreover, we adopt adaptive wavefront shaping to engineer the SPP mode, which cannot be performed using conventional plasmonic waveguides[18,19]. Figure 1e depicts adaptive wavefront shaping, which

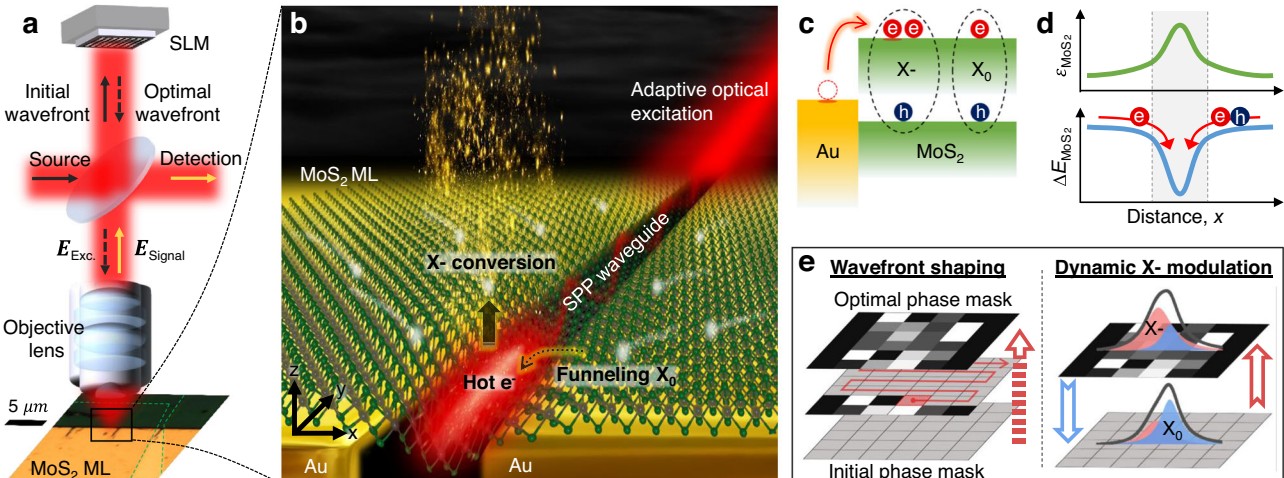

**Fig. 1 | Illustration of propagating SPP and exciton-to-trion conversion with dynamic optical modulation. a** Schematic diagram of all-optical trion control platform operating with adaptive optical excitation. Green dashed line indicates transferred $MoS_2$ ML. **b** Illustration of all-optical trion control platform facilitated by nanoscale strain gradient, plasmon-induced hot electrons, and resultant exciton-to-trion conversion. **c** Illustration of hot electron injection process from Au to $MoS_2$ ML in SPP mode. **d** Strain ($\epsilon$) and corresponding bandgap energy change ($\Delta E$) diagram of $MoS_2$ ML as function of distance $x$, where gray region indicates nanogap area. **e** Adaptive wavefront shaping using stepwise sequence feedback algorithm to find optimal phase mask (left) and dynamic excitonic emission modulation through phase mask control (right).

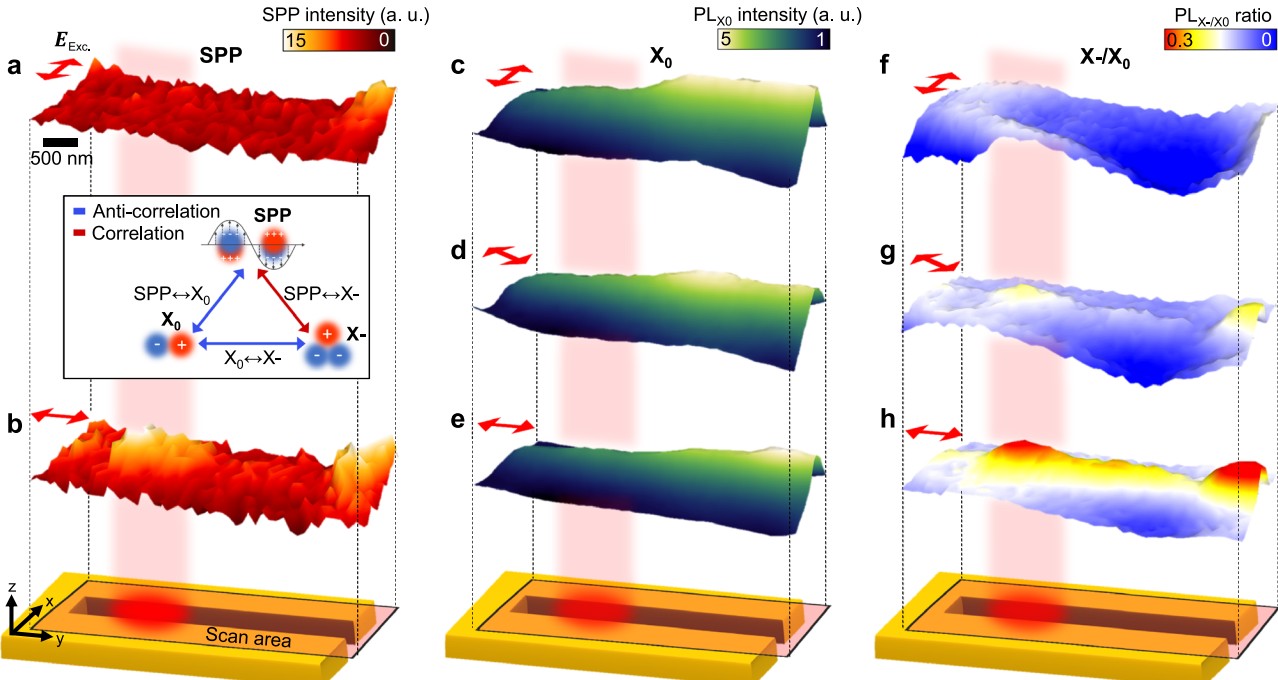

**Fig. 2 | Polarization-dependent hyperspectral imaging of SPP, $X_0$, and X- at nanoscale waveguide.** SPP images with excitation polarization across (**a**) and along (**b**) waveguide. **c–e** $X_0$ PL images with different excitation polarizations.

**f–h** PL images of X-/$X_0$ ratio with different excitation polarizations. Illustration of scan area on the waveguide is based on Rayleigh scattering image (Supplementary Figs. 4 and 5). Inset: (Anti-) correlation between SPP, $X_0$, and X-.

uses a stepwise sequence feedback algorithm to optically modulate the SPP mode of the waveguide. The optimal phase mask can significantly increase X- emission in the weak SPP region and enable instantaneous switching between the dominant $X_0$ and X- emissions.

## Polarization-dependent spatial distributions of SPP, $X_0$, and X-

We investigate the polarization-dependent activation of the SPP mode in the waveguide and its effect on $X_0$ and X- densities[21,22]. Figure 2a shows the spatial distribution of the SPP with excitation polarization across the waveguide, i.e., the waveguide is deactivated. As expected, no evidence of the SPP mode is observed in the waveguide. However, when the excitation polarization is along the waveguide, i.e., fully activated, a strong SPP mode is observed in the waveguide, as shown in Fig. 2b. The spatial distribution of $X_0$ also exhibits a polarization-dependent response. When the waveguide is deactivated, an enhanced $X_0$ density is observed at the nanogap, which is attributed to the funneling effect of the strain gradient of the nanogap, as shown in Fig. 2c. By contrast, when the waveguide is activated, the $X_0$ density near the SPP mode gradually decreases, as shown in Fig. 2d, e, where the waveguide is partially activated and fully activated, respectively. Interestingly, the spatial distribution of the X-/$X_0$ ratio exhibits an opposite behavior from that of $X_0$. When the waveguide is activated, the gradual emergence of localized X- is observed in the SPP mode, as shown in Fig. 2f–h, where the waveguide is deactivated, partially activated, and fully activated, respectively. Specifically, localization of the X- emission is observed in the SPP mode, which is attributed to the 1D strain gradient of the nanogap geometry and the SPP-induced local enhancement of the electron density. The covariance map in Fig. 2 illustrates the resulting correlations of SPP, $X_0$, and X-. It can be observed that X- is correlated with the SPP, whereas $X_0$ is anticorrelated with both the SPP and X-. This indicates a stepwise process−plasmon-induced hot electron generation, funneling of the injected electrons toward the nanogap center, and additional exciton-to-trion conversion. Note that we exclude the possible contribution from

defect-induced charges while confirming the role of lateral MIM waveguide with control experiments at low excitation power (Supplementary Figs. 6–8).

## Radiative control of trions with complete exciton-to-trion conversion

We then target the spot of the strong SPP mode and measure the time-resolved photoluminescence (TRPL) traces, as shown in Fig. 3a, b. The TRPL traces are fitted by a biexponential function with fast ($\tau_1$) and slow ($\tau_2$) components[23]. Unlike previously reported plasmon-coupled platforms, exhibiting significant decreases in decay time[24–26], both components derived from lateral MIM waveguide show minimal changes in decay time compared to the ones from silicon. Specifically, the strain gradient geometry exploits the funneling of electrons and high exciton-to-trion conversion efficiency[6], resulting in the smaller number of injected electrons to achieve complete exciton-to-trion coversion. Therefore, we induce high electron density and correspondingly enhanced trion emission while weakly coupled to the plasmon, as shown in Fig. 3b. Next, we measure spatial-dependent PL responses with three different excitation polarizations, as shown in Fig. 3c. When the waveguide is deactivated, no spectral changes are observed in the SPP mode. By contrast, when the waveguide is partially activated, emergence of the X- emission is observed in the SPP mode, although with significant emission of $X_0$. Finally, when the waveguide is fully activated, a high-purity X- emission with negligible $X_0$ emission is produced. To further investigate the polarization-dependent behavior of $X_0$ and X-, we fit the PL spectra measured at the center of the SPP mode to the Lorentz function, as shown in Fig. 3d (Supplementary Fig. 9). With the waveguide deactivated, a dominant $X_0$ emission is observed with the X- shoulder, indicating low electron density at the strain gradient center. We note that the intrinsic X- emission at the deactivated waveguide originates from the funneling of the background electrons at the strain gradient and the intrinsic polarization ratio of the excitation source (100:1). With the waveguide partially activated, an additional SPP-mediated exciton-to-trion conversion is

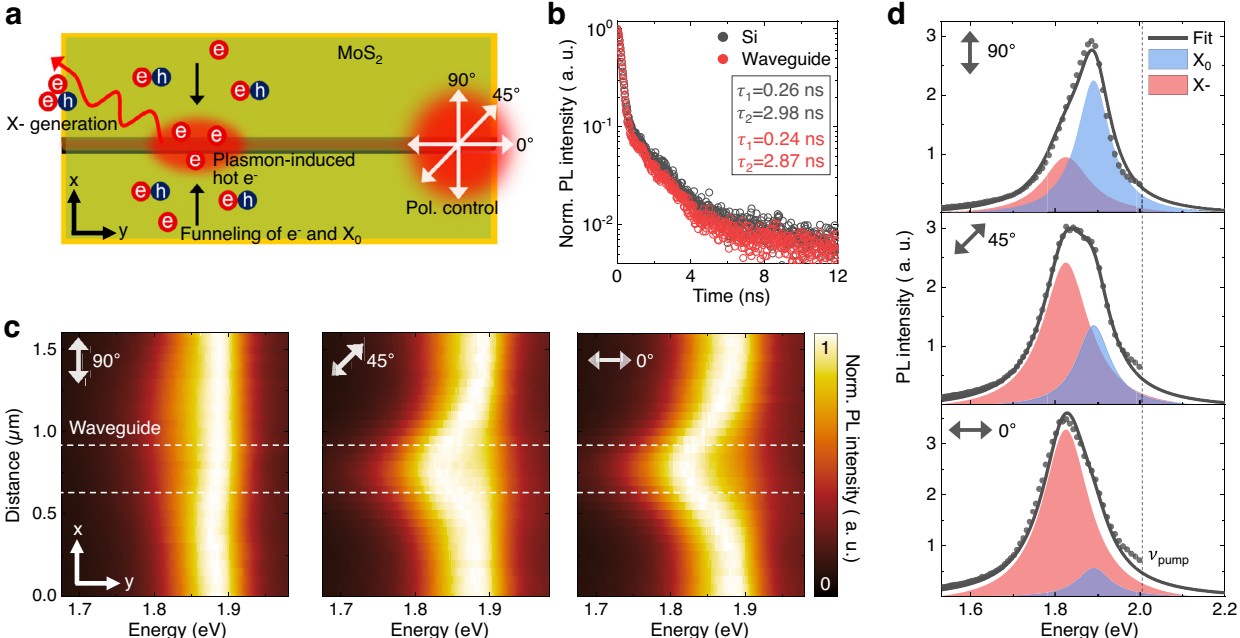

**Fig. 3 | Complete exciton-to-trion conversion. a** Illustration of exciton-to-trion conversion process assisted by plasmon-induced hot electrons. **b** Normalized TRPL traces of MoS$_2$ monolayer on silicon (black) and waveguide structure (red). **c** Spatial dependent PL spectra obtained by vertically crossing waveguide (white dashed line) with different excitation polarizations. **d** Corresponding PL spectra obtained at center of SPP mode fitted to Lorentz function.

observed, i.e., a decrease in the X$_0$ emission in contrast to an increase in the X- emission. Finally, with the waveguide fully activated, a highly dominant X- emission is produced, indicating that high-purity localized X- is achieved via a complete exciton-to-trion conversion (Supplementary Figs. 10 and 11). The minor X$_0$ portion of the emission is attributed to the diffraction-limited beam size (~450 nm), which exceeds the nanogap size (~300 nm). These results indicate the polarization-controllable exciton-to-trion conversion ratio and the dynamic transition between the high-purity localized X- state and the dominant X$_0$ state.

## All-optical spatio-spectral modulation of exciton–trion interconversion

Conventional static plasmonic waveguides have limitations in providing dynamic spatial controllability for the exciton-to-trion conversion region owing to their fixed SPP mode. To further enhance device expandability, deterministic spatial control of the SPP mode is highly desirable. To implement this, we use adaptive wavefront shaping with the SLM, as illustrated in Fig. 4a. We move the detection spot to the weak SPP region and implement a sequence feedback algorithm, which aims to maximize the target intensity by optimizing the wavefront (Supplementary Figs. 12 and 13)[18,19]. Figure 4b shows the evolution of the plasmon intensity. The plasmon intensity gradually increases during the wavefront shaping and consequently reaches an enhancement of ~210% with the optimized phase mask, as shown in Fig. 4c. This trend implies that the SPP mode can be spatially modulated at the desired location, enabling the spatial modulation of the exciton-to-trion conversion region. We then compare the PL spectra with and without the optimal phase mask, as shown in Fig. 4d. Without the optimal phase mask, the PL spectrum exhibits a dominant X$_0$ emission because of a lack of electrons, as expected. By contrast, when the optimal phase mask is used, X- emission becomes dominant, owing to the provision of extra electrons by plasmon-induced hot electron generation. We note that the increase in X- intensity is higher than the decrease in X$_0$ intensity with the optimal phase mask. This is due to the SPP-induced excitation of additional X$_0$, consequently converted to X-

(Supplementary Figs. 10 and 11). This fully optical process offers non-invasive modulation with excellent repeatability. This allows instantaneous exciton-to-trion conversions at desired locations, enabling dynamic switching between the dominant X$_0$ and X- emissions, as indicated by the two spectra in Fig. 4d–f (Supplementary Figs. 14 and 15).

## Theoretical investigation of plasmo-excitonic transport and conversion dynamics

We analyze the drift-diffusion model using experimentally obtained Kelvin probe force microscopy (KPFM) data to investigate the physical mechanism of electron funneling and the related exciton-to-trion conversion dynamics. The movement of electrons at the nanogap of the waveguide can be experimentally estimated from the work function image (Supplementary Fig. 16). Note that work function $\varphi = E_{vac} - E_F$, where $E_{vac}$ is the vacuum level and $E_F$ is the Fermi level. As shown in Fig. 5a, b, an increase in the work function is observed in the gradient region of tensile strain in the nanogap, which is in good agreement with the results of a previous study[27]. Because the gradual increase in the work function shown in Fig. 5c reaches its maximum at the nanogap center, the plasmon-induced hot electrons at the interface of Au and MoS$_2$ ML in the SPP mode can be funneled into the nanogap center. As a subsequent step, we theoretically estimate the spatial distribution of the electron and X$_0$ in the presence of the nanoscale strain gradient. First, we obtain the fitted line-shape function based on the topography profile of the MoS$_2$ ML suspended on the nanogap, as shown in Fig. 5d (Supplementary Fig. 17). The spatial distribution of the photoexcited excitons $n(x)$ can be derived by solving the steady-state continuity condition for the excitonic diffusion and drift currents, as follows:

$$\nabla(D\nabla n(x)) + \nabla(\mu n(x)\nabla u(x)) - \frac{n(x)}{\tau} - n^2(x)R_A + S(x) = 0, \quad (1)$$

where $D\nabla n(x)$ is the excitonic diffusion current term, $\mu n(x)\nabla u(x)$ is the excitonic drift current term[6,28], $D$ is the diffusion coefficient,

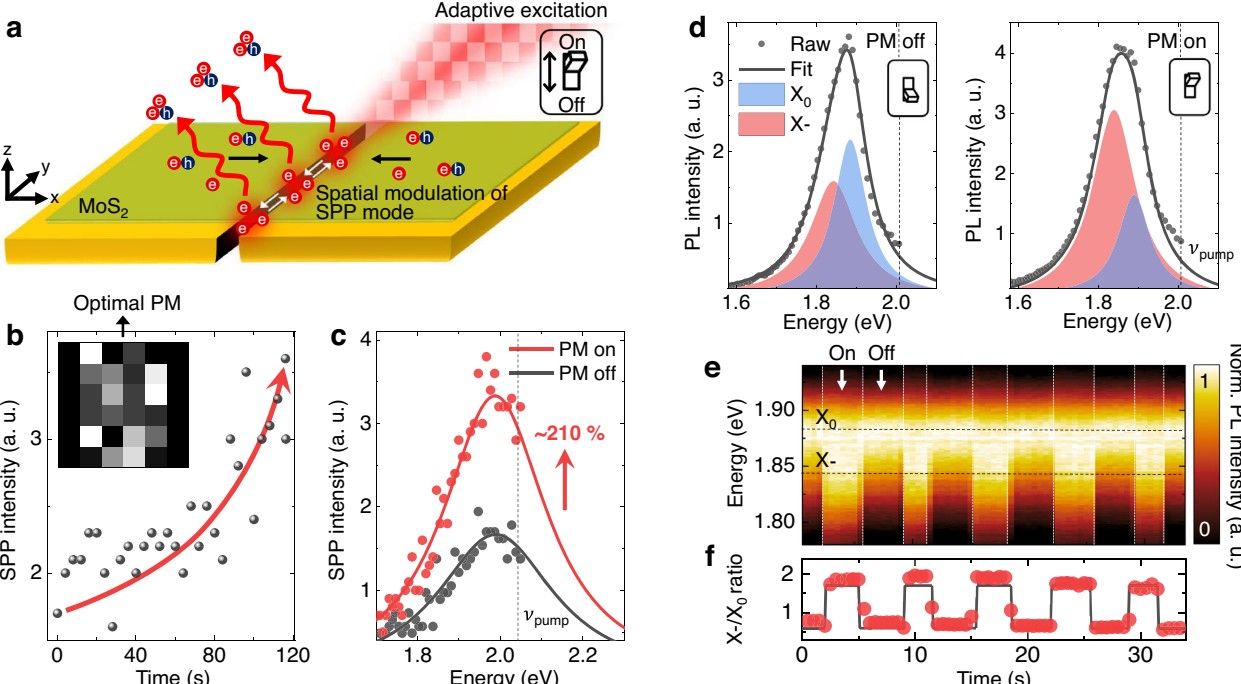

**Fig. 4 | All-optical control of exciton–trion interconversion. a** Illustration of spatio-spectral modulation of SPP mode and excitonic emission response through adaptive wavefront shaping. **b** Evolution of SPP intensity during stepwise sequence feedback. Inset: optimized phase mask (PM) after sequence feedback. **c** SPP spectra before (black) and after (red) wavefront shaping. **d** Corresponding PL spectra before (left) and after (right) wavefront shaping fitted to Voigt function. Black, blue, and red lines indicate fit of raw spectrum, $X_0$, and X-, respectively. Black dots indicate raw data. **e** Time-series normalized PL spectra during on/off switching of optimal phase mask obtained in (**b**). **f** Corresponding X-/$X_0$ ratio.

$u(x) = E_g - 0.05\varepsilon(x)$ is the strain-induced bandgap ($E_g$) change under the strain profile $\varepsilon(x)$ of MoS$_2$ ML[29], $\mu = D/k_BT$ is the mobility ($k_B$ is the Boltzmann constant, and $T$ is the temperature), $R_A$ is the Auger recombination rate, $\tau$ is the exciton lifetime, and $S(x) = \frac{I_0}{2\pi\sigma^2}e^{-x^2/2\sigma^2}$ is the exciton generation rate in a Gaussian illumination profile ($I_0$ is the intensity and $\sigma = FWHM/2\sqrt{2\ln 2}$). The diffusion coefficient values, exciton lifetime, and Auger recombination rate of the MoS$_2$ ML are obtained from previous studies[30–32]. We note that, at this stage, the photoexcited exciton density $n(x)$ includes all kinds of photoexcited excitons, e.g., neutral excitons ($X_0$) and charged excitons (X-). The spatial distribution of the electron $n_e(x)$ can be described based on an assumption of the absence of electron generation by illumination ($S(x) = 0$, $\tau = 0$, and $R_A = 0$). This yields

$$n_e(x) = \frac{N_0 e^{\nabla u_c(x)/k_BT}}{\int e^{\nabla u_c(x)/k_BT}xdx},\qquad(2)$$

where $N_0$ is the number of electrons in the entire area, and $\nabla u_c(x)$ is the strain-induced change in the conduction band[6]. Figure 5e shows the calculated profiles of the photoexcited exciton and electrons densities. The experimental results indicate that the electron density increases at the center of the nanogap, as expected. Consequently, the electron density decreases in the vicinity of the nanogap because of the funneling of background electrons toward the nanogap center. Meanwhile, with regard to the density of the photoexcited excitons, a similar tendency is exhibited; the photoexcited exciton density is funneled toward the nanogap center. Note that we subtract the photoexcited exciton density obtained without the strain gradient from the photoexcited exciton density with the strain gradient to clearly demonstrate the effect of strain gradient and consequently evaluate the density of drifted photoexcited excitons while excluding the effect of optical excitation (Supplementary Fig. 18).

The term $\frac{N_0}{\int e^{\nabla u_c(r)/k_BT}xdx}$ in $n_e(x)$ can be considered as an integration constant, which is related to the global defect density of the sample[6]. If we define this global defect density of the sample as $\alpha$, then an increasing $\alpha$ indicates the provision of extra electrons, i.e., $\alpha$ is proportional to the electron density $n_e(x)$. In our experiment, increasing $\alpha$ can be realized by plasmon-induced hot electrons, as it increases the background electron density near the nanogap. Subsequently, we gradually increase $\alpha$ and plot the evolution of the spatial distribution of the electrons $n_e(x)$ to investigate the electron density at the nanogap center when plasmon-induced hot electron generation occurs in the SPP mode. With increased background electron density, a significant increase in the electron density is observed at the nanogap center, as shown in Fig. 5f. In this case, we now consider the contribution of X- because the actual solution of drift-diffusion model $n(x)$ is the summation of $X_0$ and X- densities, i.e., $n(x) = n_{ex}(x) + n_{tr}(x)$. To include this, we use the mass action model, which is expressed as follows:

$$n_{tr}(x) = \frac{n(x) + n_e(x) + n_A(x) - \sqrt{(n(x) + n_e(x) + n_A(x))^2 - 4n(x)n_e(x)}}{2},\qquad(3)$$

where $n_A(x) = \frac{4m_{ex}m_e}{\pi\hbar m_{tr}}k_BTe^{-E_T/k_BT}$ ($m_{ex}$, $m_{tr}$, and $m_e$ are the masses of $X_0$, X-, and electron) is the relation of connecting concentrations of $X_0$, X-, and electrons by the law of mass action[6,33,34]. Figure 5g shows $X_0$ and X- densities at the nanogap center as functions of $\alpha$. As expected, the X- density is zero for $\alpha = 0$, i.e., there are no electrons; however, it continuously increases as $\alpha$ increases. Conversely, a decrease in the $X_0$ density effectively proves the occurrence of the exciton-to-trion conversion process. Finally, we compare the experimental results from the polarization control (Fig. 3) and SLM control (Fig. 4) to the theoretically analyzed data. Specifically, we derive the X-/$X_0$ ratio from

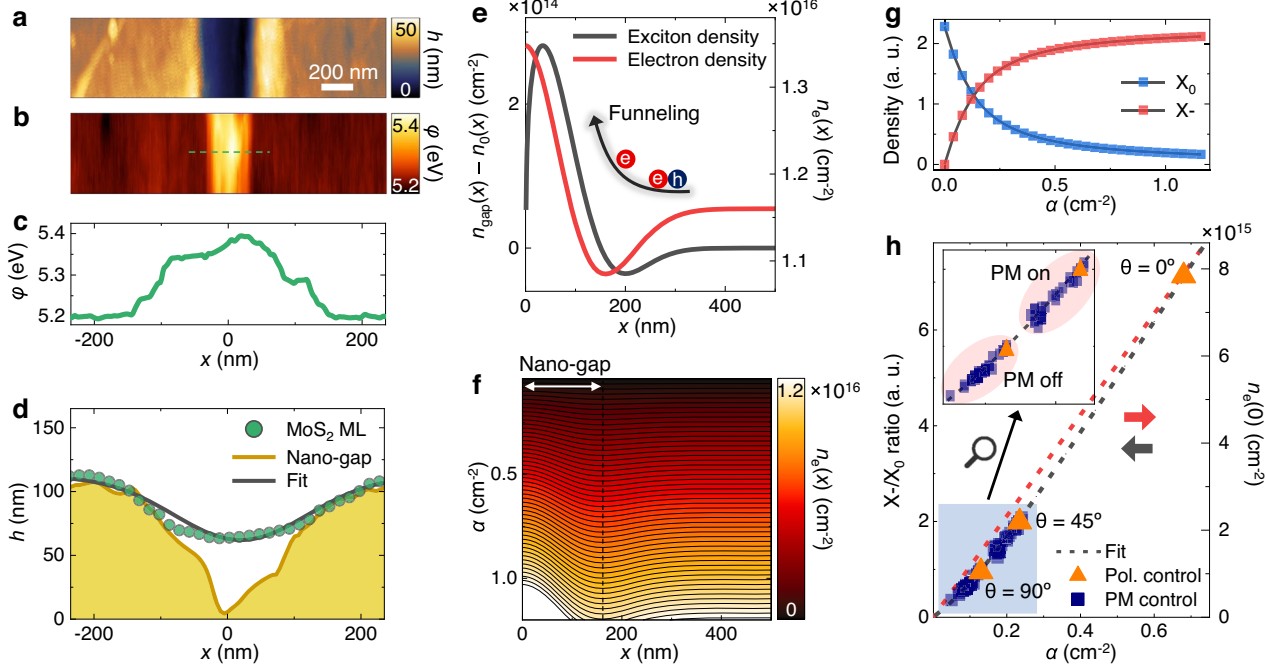

**Fig. 5 | Theoretical investigation of electron funneling and exciton-to-trion conversion dynamics.** Topography (**a**) and work function ($\varphi$) images (**b**) of waveguide obtained by KPFM. **c** Work function profile derived from dashed green line in (**b**). **d** Height profile of $MoS_2$ ML on nanogap of waveguide (green dots), fitted line shape function (black line), and height profile of nanogap without $MoS_2$ ML (yellow line). **e** Spatial density distribution of photoexcited excitons (black) and electrons (red) under strain profile estimated from fitted line-shape function in (**d**). **f** Spatial electron density distribution as function of global defect density $\alpha$ for estimated strain profile. **g** $X_0$ density (blue) and X- (red) density as functions of $\alpha$. **h** X-/$X_0$ ratio as function of $\alpha$. Dashed black line represents theoretically obtained fit from (**g**). Dashed red line indicates theoretically driven electron density at center of nanogap, i.e., $n_e(0)$ as function of $\alpha$. Orange triangles and navy squares indicate experimentally obtained values from Figs. 3 and 4, respectively. Inset: closed view of blue-filled region; values at bottom left are without optimal phase mask, whereas those at top right are with optimal phase mask.

the experimental results and match the corresponding $\alpha$ value to estimate the experimental electron density localized at the nanogap center in the SPP mode. As shown in Fig. 5h, the X-/$X_0$ ratio from the polarization control exhibits a positive correlation with the electron density. By assuming ~100% exciton-to-trion conversion efficiency under the strain gradient[6], we quantitatively estimate the electron density based on the experimentally obtained X-/$X_0$ ratio, exhibiting a maximum ten-fold enhancement of the localized electron density[35]. Similarly, the SLM-controlled X-/$X_0$ ratio exhibits two distinct regions assigned to the X-/$X_0$ ratio with and without the optimal phase mask. Therefore, we confirm the local enhancement and confinement of plasmon-induced hot electrons at the nanogap center in the SPP mode and its effect on the additional exciton-to-trion conversion process.

## Discussion

We developed an adaptive waveguide platform that enables the generation of high-purity trions, dynamic exciton–trion interconversion, and their spatial modulation in 2D semiconductors. Specifically, we showed the precise controllability of the exciton-to-trion conversion rate, which enables a dynamic transition between the dominant $X_0$ state and high-purity X- state via the modulation of the excitation polarization. Furthermore, the spatial controllability of the SPP mode was facilitated with adaptive wavefront shaping by the SLM, leading to deterministic positioning of the exciton-to-trion conversion region. Exploiting the drift-dominant exciton flux and converting confined excitons to trions through the nanoscale strain gradient result in the efficient harvesting of excitonic quasiparticles in the form of trions. Unlike highly radiative trions in plasmonic cavity platforms, our high-purity trions exhibit their intrinsic temporal characteristics, leading to the high-efficiency photoconversion[36,37]. Meanwhile, generating trionic

flux with the converted trion should be a pressing matter, as it opens a pathway toward manipulating excitonic/trionic flux efficiently at the nanoscale combined with existing plamonic MIM waveguide geometry[38–40].

## Methods

### Fabrication of nanogaps through focused ion-beam milling

Silicon wafers with thermally grown $SiO_2$ and a thickness of 1 μm were purchased from University Wafers, Boston, USA. Electron-beam evaporation was used to deposit 150-nm-thick Au on the wafers. An FEI Nova 600 dual-beam system was used to perform focused ion beam (FIB) milling on the wafers to etch into the Au and silica layers, creating a nanogap. This part of the procedure was performed at an ion beam voltage of 30 kV and current of 10 pA. This Au layer was then removed using gold etchant (TFA, Transene Company Inc.), and a fresh layer of Au (50 nm) was deposited onto the wafer using e-beam evaporation. FIB milling was performed again at voltage 30 kV and a lower current (1 pA) to selectively etch the Au from the bottom of the nanogaps[41].

### Growth and transfer of $MoS_2$ MLs

A two-zone furnace was used to grow the $MoS_2$ ML flakes; sulfur flakes (Merck, ≥99.99%) were placed in the upstream zone; a 0.01 M sodium molybdate aqueous solution was spun onto a $SiO_2$/Si substrate as the molybdenum precursor; the substrate was loaded into the downstream zone; the sulfur flakes and substrate were heated at 200 °C and 750 °C temperatures, respectively, for 7 min and maintained for 8 min; the substrate was then cooled naturally to room temperature. The entire process was performed with a $N_2$ carrier gas at a flow rate of 600 SCCM. The as-grown $MoS_2$ was then coated with poly (methyl methacrylate) (PMMA) at 2500 rpm for 1 min. To delaminate the $SiO_2$/

Si substrate, the PMMA-coated sample was floated on a 2 M aqueous KOH solution. After delamination, the KOH residues were rinsed several times with deionized water. The PMMA/$MoS_2$ layer was scooped with a nanogap-patterned substrate. Finally, the PMMA layer was removed using acetone and isopropyl alcohol (IPA).

## Photoluminescence spectroscopy setup

The prepared TMD MLs on the nanogap were loaded onto a piezo-electric transducer (PZT, P-611.3X, Physik Instrumente) for XY scanning. To obtain a high-quality wavefront of the excitation beam, a He–Ne laser (594.5 nm, <1.0 mW) was coupled and passed through a single-mode fiber (core diameter of ~3.5 μm) and collimated again using an aspheric lens. Finally, the beam was focused onto the sample using a microscope objective (NA = 0.8, LMPLFLN100X, Olympus). The PL responses were collected using the same microscope objective (backscattering geometry) and passed through an edge filter (FEL0550, Thorlabs) to remove the fundamental laser line. The PL signals were then dispersed onto a spectrometer (f = 328 mm, Kymera 328i, Andor) and imaged using a thermoelectrically cooled charge-coupled device (CCD, iDus 420, Andor) to acquire the PL spectra. Before the experiment, the spectrometer was calibrated using a mercury–argon lamp. A 150 g/mm grating blazed to 800 nm (spectral resolution of 0.62 nm) was used for PL measurements. Time-resolved PL measurements were performed with a time-correlated single-photon counting (TCSPC) method. A commercially available TCSPC module (PicoHarp, PicoQuant GmbH) was used to obtain the PL decay curves. A 405 nm picosecond laser diode with an 80 MHz repetition rate was used as an excitation source.

## Adaptive wavefront shaping

To manipulate the SPP mode of the plasmonic waveguide, a recently developed adaptive optics technique, i.e., wavefront shaping of the excitation beam, was used. For wavefront shaping, a feedback loop was made using a simple stepwise sequential algorithm[18]. Specifically, 600 × 600 pixels of the SLM active area with liquid crystals were divided into 6 × 6 segments. Each segment swept its phase from 0 to 2π to determine the optimal phase providing the strongest target signal, i.e., the plasmon intensity. An optimized phase mask was obtained via repetition of this feedback algorithm for all segments.

## Kelvin probe force microscopy

We measured the work function of the region of a gap (suspended ML $MoS_2$) sample by using a conventional AFM (Park Systems Co., NX10) with a measurement mode of Kelvin probe force microscope (KPFM) addressing nanoscale surface work function. We used a commercial Cr & Au coated cantilever (Mikromasch Co., NSC18). Before the main measurement, we performed a calibration on the work function of cantilever by scanning the highly ordered pyrolytic graphite (HOPG, ~4.6 eV), commonly used as calibration method for KPFM. Then, we performed the nanoscale work function mapping on the suspended ML $MoS_2$ with the calibrated cantilever showing maximum value of about 5.4 eV. Note that the KPFM measurement was performed without optical excitation to precisely characterize the work function of $MoS_2$ ML without external perturbation.

## Data availability

The data that support the plots within this paper and other findings of this study are available from the corresponding author upon reasonable request.

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

## Acknowledgements
This work was supported by the projects of 2020R1C1C101130113, (2022) ERIC_03_3, IITP-2022-RS-2022-00164799, ETRI_21YB2100, and 2021R1A6A1A10042944. We gratefully acknowledge the critical support and infrastructure provided for the fabrication of the waveguide by the Kavli Nanoscience Institute at Caltech and Samsung Advanced Institute of Technology. S.A. and H.S.L. acknowledge National Research Foundation of Korea (NRF) grant funded by the Korea government (MSIT) (2022R1A4A1033358). H.S.L. acknowledges 2021R1A2C1003074. S.H.C. and K.K.K. acknowledge the support by the Institute for Basic Science (IBS-R001-D1) and Advanced Facility Center for Quantum Technology. K.K.K. acknowledges the Basic Research Program (2020R1A4A3079710, 2022R1A2C2091475) and the Next-generation Intelligence Semiconductor Program (2022M3F3A2A01072215) through the National Research Foundation of Korea (NRF) funded by the Ministry of Science and ICT.

## Author contributions
H.L. and K.-D.P. conceived the experiments. H.L. performed PL spectroscopy and control experiments. H.L. and Y.K. performed the adaptive wavefront shaping experiments. S.K., R.H.S., and H.C. designed and fabricated the nanogap-based waveguide devices. H.L. and H.J. performed the theoretical calculations and modeling by analyzing the exciton-diffusion model. Y.J. and S.A. obtained the KPFM images. D.G.H. and H.S.L. performed the time-resolved PL measurements. S.H.C. and K.K.K. prepared and transferred MoS₂ ML onto the nanogap-based waveguide device. H.L., Y.K., M.K., and K.-D.P. analyzed the data, and all authors discussed the results. H.L. and K.-D.P. wrote the manuscript with contributions from all authors. K.-D.P. supervised the project.

## Competing interests
The authors declare no competing interests.
