## [Peer Review File · Nature Communications]

Reviewers' Comments:

Reviewer #1:

Remarks to the Author:

The manuscript presented the method to control the exciton-to-trion conversion process in a monolayer MoS₂ via inducing a nanoscale strain gradient in a 2D crystal transferred on a metal-insulator-metal waveguide at room temperature. Basically, the hot electrons generated by the metal have provided the excess electron, which is transferred to the MoS₂ and is responsible for the control of the trion. I think the technical presentation of the paper is good. However, there are various significant queries regarding quasi-particles that need to explain, which lack to be discussed in the present form of the manuscript. The following questions are essential to be addressed to make the manuscript more clearly understood.

Query 1. The author should discuss the advantage of the trions and exciton-based applications and also can mention how the trionic devices affect the substrates (comparing SiO₂/Si, SiN/Si, Metallic layer or pillar etc.). As the author's cited references 3 and 4 for the claim for "various exciton-based optoelectronic devices," which are not suitable. Both the references only discussed how to control the excitons or trions via strain, instead of showing direct optoelectronic excitonic applications.

Query 2. In the existing literature, there are demonstrating the various way to control the excitonic and trionic properties (except electrical and chemical), such as tuning the excitation energy (J. Phys. Chem. C 2021, 125, 32, 17806–17819), formation of pico cavity (Sci. Adv. 2019;5: eaau8763), ion or laser-induced defect, formation of heterostructure (ACS Appl. Mater. Interfaces 2022, 14, 33482-33490), etc. Furthermore, various reports have discussed the metal plasmon implantation on monolayer MX₂ or WX₂ for well controlling the quasiparticle behavior at room temperature (Physica E 140 (2022) 115213). So, what is so special when choosing the MIM substrate for controlling the exciton-to-trion population? Which one is better (metal plasmon induced by nanocavity/particle or MIM substrate) and why? Although the authors have mentioned that at ambient conditions, ML surface significantly reduces electron density, and that is why the enhancement of quasiparticle density through other technique is less, the cited literature (ref 10 and 11) have not directly marked this statement. However, ref 11 directly exposed the O₂ and H₂O, and it claimed the reduction of PL spectra (essential to note this point), not due to the atmospheric oxygen (as the authors said). Justify it in brief.

On the other hand, during the transfer, it can also be possible might have created a lot of defects that will change the concentration/density of quasiparticles (exciton or trion). In the present case, the authors transferred the monolayer CVD-grown MoS₂ to the MIM substrate. Have the authors done the XPS/HRTEM before and after transferring the monolayer for confirming the trion enhancement due to MIM, not due to defects?

Query 3. The authors previous paper has reported similar physics (Sci. Adv. 8, eabm5236 (2022)) on the Au nanogap (or suspended ML MoS₂). The design of the layer structure is the same as that of the waveguide (present case). As a reader, I could not understand why the surface plasmon polariton (SPP) mode of the plasmonic has not been activated in previously published layer structure (Au nanogap device, Sci. Adv. 8, eabm5236 (2022)). The authors should explain the reason and also add a few lines in the introduction to clarify the difference between both works.

Query 4. Figure S2 (supplementary information) couldn't directly justify the following statement "decrease the electron density at the nanogap." For that author should do a similar measurement at three different conditions- air (ambient), vacuum, and oxygen environment and then compare the results for claiming the decrease in electron density in the presence of an oxygen atmosphere. The PL intensity (trion increasing more faster than exciton) always increases with laser power. Even if you decrease the power to < 0.1 mW, the defect states generate a large number of defect electrons near the Fermi level, which will be capable of forming trions (In the present case, power is very large > 0.1 mW). Did you measure the conversion efficiency (exciton-to-trion) at a low power range (1 μ W to 20 μ W) because, at these low powers, the defect electrons are less active, and enhancement of trions may occur entirely due to plasmons?

The authors find another way to justify the statement of "decreased electron density in a nanogap." Did the authors measure the PL in a vacuum? How many times transferred ML MoS₂ on the MIM substrate repeat the measurement? How much gap in an SPP waveguide and Au nanogap in a previous paper (Sci. Adv. 8, eabm5236 (2022)).

Query 5. In Figure 3 d, the authors only chose two peaks for fitted the PL spectrum at different polarization. Are there any defects and bi-excitons peaks that have been observed? I believe that

at higher power ($> 0.1\text{mW}$), defects and bi-excitons (might be) peaks are usually obtained because CVD-grown films always present defects states, and here you transferred CVD ML to another substrate but not getting any defects peaks in any of the power (strange). The authors must in clarifying this issue. What laser power was kept during the measurement of figure 3d data? Was the recorded PL spectrum checked in a vacuum?

Query 6. Did the authors measure the degree of polarization (DOP) of both samples (with and without transfer to MIM)? If not, I suggest doing this for more understanding of the following polarization-dependent exciton-to-trion enhancement phenomena.

In Figure 3d, the FWHM of the PL peaks varied a lot from 0° to 90° , justifying the FWHM changes. At 0° polarizing angle, the FWHM is too broad compared with 45° . Why? I suggest authors further check the PL spectra fitting parameters and consider the defects or bi-excitons (or others justify peaks, if available) in a given PL spectra.

Query 7. The objective of the present manuscript is to enhance the trion signal/density (by reducing exciton) in a PL spectrum, so the most essential question, enhancement of trions, are singlet or triplet? I suggest that the authors should discuss a few lines in the text about the type of trions. How are the hot electrons affecting the transition at K and K'? Do authors have any phenomenal model for explaining the hot electron (or SPP mode) coupling with excitons (for forming trions) at K and K' (interband and intraband transition roles)?

Phonon plays an important role in the formation of trions. Is there any exciton-phonon interaction occurring for the coupling of SPP electrons for the making of trions (I mean, the picture is not so simple that all-time excitons couple with SPP electrons and form trions)? Could any Fermi level affecting while interacting with the plasmon signal? If yes, explain.

Query 8. Has the phase mask been implemented via SLM incorporated with light? The SPP intensity has been measured along with PL intensity (setup shown in Fig. 1 a). If yes, then how to distinguish the signal because both the emissions (ML PL and SPP) are in the same range?

Query 9. The ML WS₂ on to Si/SiO₂ substrate at low temperature ($<77\text{ K}$) becomes an excitonic rich in nature (means: more excitons and less or no trions), so what authors believe that the proposed conversion scheme would work in this case? The author did observe a similar MIM scheme for other similar MX₂/WX₂ types of materials (for a generalized proposed scheme robust).

Query 10. In Fig S10 a, representing $n(x)$ is a spatial distribution of the exciton with (red) and without (black) strain gradient, while in the text, authors are writing $n(x)$ is the summation of X₀ and X- spatial density distribution i.e., $n(x) = n_{ex}(x) + n_{tr}(x)$. Clarify the notification and statement.

Query 11. In the theoretical section, so many notations are not well appropriately described, such as ϕ (work function usually defined as $E_{vac}-E_F$) is the same here or different definition has been considered by authors. What are n_A and n_B , and how is it affecting the experimental data? The authors have experimentally estimated n_B (in Fig 5 e), but the objective and correlation are not clear well.

Query 12. Fig 5 a, b, and c have been estimated via KPFM, and the meaning of ϕ is the same as I defined above. If yes, so the max. value of the work function is around 5.4 in a gap (suspended ML MoS₂) but the estimated value (by others) is 6.11 eV, in slight the difference of the work functions and briefly explains the details of measurement of KPFM for more clearance of reader. During the measurement of KPFM, applying the voltage near the tip may tune the Fermi level of the material and also may create local defects, resulting producing unbound electrons that will easy to couple with excitons. May it could be the reason for enhancing the trions more coupled with the SPP plasmons? What is the vision of the authors regarding the above statement?

Is there any correlation between Fig 5 e (main text) and Fig S10 (supporting document)? Has the difference of Fig S10 data presented in Fig 5 e? I suggest the authors should rewrite the theoretical part properly (explain all the parameters properly) and make a clear bridge between theoretical and experimental results.

Reviewer #2:

Remarks to the Author:

This manuscript demonstrated an all-optical control method for the exciton-to-trion conversion and spatial distribution control in a MoS₂ monolayer. By designing nanoscale strain gradient in MoS₂ monolayer transferred on a metal-insulator-metal (MIM) waveguide, they accomplished the efficient electron density and exciton-to-trion conversion. In addition, by using adaptive wavefront

shaping, the surface plasmon polaritons (SPPs) mode can be modulated, so that an all-optical control of the exciton-to-trion conversion rate and trion distribution can be realized. The all-optical method for excitonic quasiparticle control proposed by this work can be quite useful for excitonic photoconversion. The results are original and new in the relevant research community, and have significance in terms of potential nano-photonics application. I would recommend further consideration of this manuscript for publication in Nature Communication, after authors address the following questions/comments:

1. Since there have been quite a few reports for the electric gate-controlled exciton/trion PL efficiency in terms of TMD ML materials, a general question is about the exciton-to-trion conversion rate comparison, between the electric control method such as application of an electric gate, chemical doping method, and the all-optical control method demonstrated in this work. It is better to give some comments/comparison for the electric, chemical and all-optical control methods for the trions generation.
2. Since the plasmonic MIM waveguide is utilized in this work to enable the plasmon-induced hot electron generation and injection to the MoS₂, and therefore the additional exciton-to-trion conversion. Could authors give some rough estimation about the generated and injected electron density? And correspondingly, the exciton-to-trion conversion efficiency? Though the calculated exciton and electron densities are shown in Fig. 5e & 5g, there are no given absolute values.
3. The Fig. S1 in the Supplementary Information, shows the Raman spectra of MoS₂ ML transferred on Au substrate and suspended on trionic waveguide. It would be good to have the optical reflectivity spectra for MoS₂ ML transferred on Au substrate and suspended on trionic waveguide as well. With the reflectivity and/or PL spectra comparison, could authors give some discussions/comments about the possible energy transfer between the SPP and excitons/trions of MoS₂ ML? And how this possibly may affect the exciton-to-trion conversion?
4. With the plasmonic MIM waveguide design, the funneled electrons and converted trions, most likely, are localized (in the nanogap center). Even the high exciton-to-trion conversion rate can be obtained by this method, the converted trions would have even lower diffusion length, it hardly expects any exciton/trions flux that can be useful for promising exciton-based optoelectronic devices when exciton flux is concerned. The authors may consider this issue and have some corresponding statement in their Introduction part, and the Discussion part as well.
5. The exciton-to-trion conversion rate and trion distribution control realized by the SPPs mode modulation demonstrated in this work is excellent result. This result suggests potential nano-photonics application for TMD materials integrated with a plasmonic MIM waveguide. This might be more promising than that of exciton/trions flux-based optoelectronic devices, as mentioned in the previous comment. So I would suggest the authors comment this issue by referring the other relevant works for nano structures integrated with a plasmonic MIM waveguide.

Reviewer #3:

Remarks to the Author:

The manuscript entitled " All-optical control of high-purity trions in nanoscale waveguide" by Lee et al. have investigated the generation of high-purity trions and dynamic exciton-trion interconversion by a designed metal-insulator-metal (MIM) waveguide with spatial controllability of the surface plasmon polariton (SPP) mode facilitated with spatial light modulator (SLM).

In this study, the authors investigated the monolayer MoS₂ suspended on the nanogap geometry of the plasmonic MIM waveguide which induces the strain gradient leading to the significantly increases the funneling efficiency for confining excitons to the nanogap. The concepts and related works have been published by the same group as listed in references 13 and 14. They further utilized the SPP mode of the plasmonic MIM waveguide to induce additional hot electrons to enable the injection of electrons from Au to the MoS₂. It was observed that the extra electrons are funneled toward the nanogap and locally increase the exciton-to-trion conversion. The polarization-dependent experiments are also performed to compare the trion conversion rate with respect to the magnitude of activation SPP modes. Moreover, the authors employ a SLM with adaptive wavefront shaping and sequence feedback algorithm to spatially control the trion emission intensity via locally enhance the plasmon intensity in the weak SPP region. The physical mechanism of electron funneling and the related exciton-to-trion conversion dynamics is analyzed by drift-diffusion model and experimentally obtained Kelvin probe force microscopy (KPFM) data. Theoretical estimation of the spatial distribution of the electron and X0 in the presence of the

nanoscale strain gradient supports experimental data.

The manuscript is well-written and well-organized with the step by step realizing the all optical control high-purity of tions dynamically and locally. Their scientific aim is precise. The work is significant to the field of optical control low-dimensional nanodevice applications. It is a pioneer work considering the adaptive wavefront shaping technique into optoelectronic materials. The work exploits SPP mode of the plasmonic MIM waveguide and 1D nanoscale strain gradient in suspended MoS₂ ML. It shows a better control of the ratio of dominant X0 emission and high-purity X-emission compared to other published results elsewhere important for exciton-based optoelectronic devices. The work supports the conclusions and claims. The methodology looks sound and matches with the standard of our field. The analysis is good and the theoretical estimations support the observed results. However, there are some comments/issues that need to be addressed before accepting for publication. The work meets the expected standards for the journal of Nature Communications if the mentioned concerns can be carefully addressed.

1. In the method description, the low power CW He-Ne laser (594.5 nm, <1.0 mW) was used to excite the sample. How to use a low power He-Ne laser instead of tip to activate the SPP mode? Is this the reason why have the weak SPP region? The SPP formation by laser needs to be further classified and compare with tip induced SPP mode.
2. The work focuses on the all optical control method however the optical power dependence is not considered in the experiment data. I suggest the authors perform the laser power dependence experiments to further conclude their findings if the hot electrons are highly dominant in the nanogap center.
3. A strong SPP mode is observed in the waveguide, as shown in Fig. 2b. How to measure the SPP mode in the experiment? It is helpful to readers if the authors describe the measurement of SPP modes in detail.
4. "Unlike previously reported plasmon-coupled platforms, which demonstrated dramatic decreases in lifetime, both components derived from the waveguide exhibit minimal changes in decay time compared with the ones from silicon. This is attributed to the strain gradient geometry, which can have higher electron densities even without being strongly coupled to the plasmon." Although the paragraph explains the TRPL result, it is still not straightforward to connect the relation of higher electron densities and life time of exciton or trion in this description. It should be further clarified for clear understanding.
5. The descriptions of all the figures in main text should be more elaborative. For example, nB in Fig. 5e needs to define in the main text properly.
6. The defect density α is related to nB and plays an important role for generating hot electrons and increase in background electrons. A little bit of more discussions on a) the relation between α vs nB and b) how one can experimentally manipulate defects in such systems in a controllable way, are suggested to include in the text.
7. The gap of the nanostructure (waveguide) is supposed to be SiO₂. How to understand the work function of the waveguide compared to the gold in the center as shown in Fig. 5(a) and Fig. 5(b)?
8. For the description of the fabrication of nanogaps through focused ion-beam milling, it is not clear for me that why it needs two times deposit of Au. More detail statements are needed.
9. For my understanding, the metal-insulator-metal structure in the work refers to the lateral structure. However, it is not clearly point out in the manuscript.

Dear reviewers,

We thank the reviewers for the helpful comments. With regard to the reviewers' concerns raised, we have addressed each of the comments and revised the manuscript correspondingly. The reviewer comments are in black, with our replies in blue, with revisions to our manuscript indicated in red in the point-by-point response.

Sincerely,

Hyuck Choo and Kyoung-Duck Park

=====

Reviewer #1:

The manuscript presented the method to control the exciton-to-trion conversion process in a monolayer MoS₂ via inducing a nanoscale strain gradient in a 2D crystal transferred on a metal-insulator-metal waveguide at room temperature. Basically, the hot electrons generated by the metal have provided the excess electron, which is transferred to the MoS₂ and is responsible for the control of the trion. I think the technical presentation of the paper is good. However, there are various significant queries regarding quasi-particles that need to explain, which lack to be discussed in the present form of the manuscript. The following questions are essential to be addressed to make the manuscript more clearly understood.

We thank the reviewer for the positive evaluation with the constructive comments to improve the completeness of our work. With regard to the raised concerns, we have performed additional experiments and addressed each of the comments with corresponding revisions to our manuscript.

1) The author should discuss the advantage of the trions and exciton-based applications and also can mention how the trionic devices affect the substrates (comparing SiO₂/Si, SiN/Si, Metallic layer or pillar etc.). As the author's cited references 3 and 4 for the claim for "various exciton-based optoelectronic devices," which are not suitable. Both the references only discussed how to control the excitons or trions via strain, instead of showing direct optoelectronic excitonic applications.

We thank the reviewer for pointing this out. Based upon this comment, we have revised the manuscript to i) include the advantages of trions and exciton-based applications and ii) cite the appropriate references, demonstrating direct optoelectronic and excitonic applications.

[Revised text] The spatial control of excitonic quasiparticles in 2D semiconductors has been extensively studied for the development of various exciton-based optoelectronic devices, especially facilitating intermedium of the electronic system and optical system, as well as highly efficient light-harvesting devices [*Science* **321**, 229 (2008), *Nano Letters* **21**, 43 (2020), *Nature Photonics* **3**, 577 (2009), *Science* **360**, 897 (2018)].

With regard to the effect of substrate on the trionic devices, metal-insulator-metal geometry should be necessarily considered to effectively excite the SPP mode in the nanogap. Specifically, the MIM geometry is designed to provide a high-contrast refractive index between the internal channel region and the substrate, providing a higher coupling efficiency [*Nature*

Photonics **6**, 838 (2012), *Science* **338**, 1317 (2012), *Nature communications* **11**, 2930 (2020)]. In our experiment, we use Au-SiO₂-Au geometry, having high effective refractive index and capability to stimulate plasmon-induced hot electrons. We have added this discussion with detailed description on the MIM waveguide geometry in the Supplementary Information.

[Revised text] By contrast, the proposed nanogap-based lateral MIM waveguide device with the designed SPP mode can supply extra electrons locally via plasmon-induced hot electron generation, as illustrated in Fig. 1b (see Fig. S3 in Supplementary Information).

[Added results in SI]

Fig. S3. Fabrication process of lateral MIM waveguide.

The coupling of light and SPP propagation rely on a lateral MIM geometry where the top and sidewalls of the nanogap are gold-coated, whereas bottom of the channel is SiO₂. For fabrication of the channel, we start with a SOI wafer, and deposit 150 nm of Au on it. Then we perform a FIB milling step to etch into the SiO₂. As seen in the Fig. S3, after milling the resultant nanogaps have a thin layer of gold on top, but much of the sidewall is SiO₂. This milling process results in a slight taper to the sidewalls, which is crucial for subsequent gold deposition step. The top layer of gold is then removed using a gold etchant. A fresh layer of Au (50 nm) is deposited using e-beam, and coats the top, sidewalls, and bottom of the nanogap. A second round of milling is then performed to remove the gold from bottom of the nanogap channel. As the figure illustrates, the first milling process is an oxide etch step, whereas the second milling is an Au etch step to remove gold from bottom of the channels.

The parameters for fabricating lateral MIM waveguide, such as gap size and height, are optimized in the way of maximizing the efficiency of SPP coupling to the nanogap while also minimizing losses during SPP propagation in the nanogap. Specifically, the nanogap lateral MIM geometry (Au-SiO₂-Au) has been designed to provide a high contrast between the effective refractive index inside the channel and the refractive index of the substrate ($n_{\text{SiO}_2} = 1.46$) providing a higher coupling efficiency. Furthermore, coupling of incoming laser light with lateral MIM nanogap to activate SPP mode proceeds through scattering from the edges and sidewalls of the gap. This method where light is coupled to Au-SiO₂-Au MIM nanogap through edge/tail-end illumination has been discussed in detail in previous works [*Nature Photonics* **6**, 838 (2012), *Science* **338**, 1317 (2012), *Nature communications* **11**, 2930 (2020)].

2) In the existing literature, there are demonstrating the various way to control the excitonic and trionic properties (except electrical and chemical), such as tuning the excitation energy (J. Phys. Chem. C 2021, 125, 32, 17806–17819), formation of pico cavity (Sci. Adv. 2019;5: eaau8763), ion or laser-induced defect, formation of heterostructure (ACS Appl. Mater.

Interfaces 2022, 14, 33482-33490), etc. Furthermore, various reports have discussed the metal plasmon implantation on monolayer MX₂ or WX₂ for well controlling the quasiparticle behavior at room temperature (Physica E 140 (2022) 115213). So, what is so special when choosing the MIM substrate for controlling the exciton-to-trion population? Which one is better (metal plasmon induced by nanocavity/particle or MIM substrate) and why? Although the authors have mentioned that at ambient conditions, ML surface significantly reduces electron density, and that is why the enhancement of quasiparticle density through other technique is less, the cited literature (ref 10 and 11) have not directly marked this statement. However, ref 11 directly exposed the O₂ and H₂O, and it claimed the reduction of PL spectra (essential to note this point), not due to the atmospheric oxygen (as the authors said). Justify it in brief.

We thank the reviewer for the helpful comment. To exploit trions as carriers in circuit and energy-harvesting systems, generating long-lived and high-purity trions and confining their spatial distribution at the desired location are highly desirable. In this light, employing MIM substrate has advantages over the large area manipulation without spatial controllability [*J. Phys. Chem. C* **125**, 17806 (2021), *ACS Appl. Mater. Interfaces* **14**, 33482 (2022)] and the metal plasmonic implantation resulting in the dramatic decrease of trion lifetime [*Physica E* **140**, 115213 (2022)]. In addition, Harats et al. [*Nature Photonics* **14**, 324 (2020)] recently revealed that the efficiency of the exciton-to-trion conversion can reach ~100 % with strain gradient geometry. Therefore, in contrast to the other methods possessing lower spatial overlap between the excitons and electrons, our all-optical control modality with strain gradient can have higher exciton-to-trion conversion efficiency, requiring a lower doping level to achieve high trion density.

With regard to the use of terminology “atmospheric oxygen,” it was inappropriately used and should be revised to “H₂O and O₂ molecules.” We have thoroughly reflected this revised terminology to the manuscript.

[Revised text] However, at ambient conditions, H₂O and O₂ molecules physisorbed onto the TMD ML surface significantly reduce electron density [*Physical Review B* **99**, 121201 (2019), *Nano Letters* **13**, 2831 (2013)].

[Revised text] However, at ambient conditions, the electron density of the MoS₂ ML noticeably decreases owing to the presence of H₂O and O₂ molecules physisorbed onto the MoS₂ ML surface [*Physical Review B* **99**, 121201 (2019), *Nano Letters* **13**, 2831 (2013)].

On the other hand, during the transfer, it can also be possible might have created a lot of defects that will change the concentration/density of quasiparticles (exciton or trion). In the present case, the authors transferred the monolayer CVD-grown MoS₂ to the MIM substrate. Have the authors done the XPS/HRTEM before and after transferring the monolayer for confirming the trion enhancement due to MIM, not due to defects?

We have performed additional experiments to investigate the defect-related charge distribution and confirm the role of MIM substrate to enhance the trion density. We have added the experimental results and the corresponding discussion in the Supplementary Information, as well as a brief statement in the main text as follows:

[Added text] Note that we exclude the possible contribution from defect-induced charges while confirming the role of lateral MIM waveguide with control experiments at low excitation power (see Fig. S6-8 in Supplementary Information).

[Added results in SI]

Fig. S6. XPS spectra of as-grown (a) and transferred (b) MoS₂ monolayer.

To exclude the defect-related electron generation, we perform X-ray photoelectron spectroscopy (XPS) before and after transferring MoS₂ monolayer onto the target substrate. As a result, we obtain Mo:S ratio of 1:1.976 for as-grown MoS₂ monolayer (Fig. S6a) while 1:1.998 for transferred MoS₂ monolayer (Fig. S6b). After the transfer of MoS₂ onto the substrate, the Mo:S ratio is highly close to 1:2, indicating negligible generation of defects during the transfer. Note that slightly dominant detection of Mo can be attributed to the Molybdenum precursor.

Fig. S7. Charge density distribution in MoS₂ ML transferred onto Au substrate. (a) A_{1g} center versus E_{2g} center measured at various spots. Probability histogram of E_{2g} center (b) and A_{1g} center (c).

To investigate the charge distribution of MoS₂ ML, we obtain Raman spectra at various spots and extract the peak position of E_{2g} and A_{1g} peak, as shown in Fig. S7a. Fig. S7b-c shows the probability distribution of E_{2g} and A_{1g} peak, respectively. The linewidth of the fitted probability distribution exhibits ~0.2 cm⁻¹, demonstrating the homogeneity of the charge distribution [*Nano Letters* **13**, 5944 (2013), *Sci. Technol. Adv. Mater.* **16** 035009 (2016)].

Fig. S8. Power dependence of exciton-to-trion conversion. (a) Excitation-power-dependent PL spectra at Au substrate. (b) Extracted X_0 intensity (blue) and X^- intensity (red). (c) Extracted X^- intensity at low excitation powers. (d) Excitation-power-dependent PL spectra at SPP mode. (e) Extracted X_0 intensity (blue) and X^- intensity (red). (f) Extracted X^- intensity at low excitation powers.

To confirm the role of SPP mode on the exciton-to-trion conversion, while excluding the effect of defect-related provision of electrons, we measure excitation-power-dependent PL spectra at Au substrate and SPP mode. At Au substrate, the linewidth of the PL spectra increases as increasing the excitation power, attributed to the emerging trion peak, as shown in Fig. S8a. In contrast to gradually increased X_0 intensity as a function of excitation power, the X^- intensity shows negligible changes at the low excitation power ($<10 \mu\text{W}$), which is possibly due to the inactivation of defect-induced electrons at low excitation power (Fig. S8b-c) [*Nano Letters* **19**, 6299 (2019)]. On the other hand, the dominant X^- peak are continuously observed with increasing excitation power at the SPP mode, as shown in Fig. S8d. Correspondingly, the X^- intensity linearly increases as increasing excitation power even at the low excitation power, as shown in Fig. S8e-f. This behavior well indicates the plasmon-induced hot electron injection [*J. Phys. Chem. Lett.* **8**, 4925 (2017)] and consequently increased X^- intensity with high exciton-to-trion conversion efficiency [*Nature photonics* **14**, 324 (2020)].

3) The authors previous paper has reported similar physics (Sci. Adv. 8, eabm5236 (2022)) on the Au nanogap (or suspended ML MoS₂). The design of the layer structure is the same as that of the waveguide (present case). As a reader, I could not understand why the surface plasmon polariton (SPP) mode of the plasmonic has not been activated in previously published layer structure (Au nanogap device, Sci. Adv. 8, eabm5236 (2022)). The authors should explain the reason and also add a few lines in the introduction to clarify the difference between both works.

To effectively activate the SPP mode in the MIM waveguide, various parameter should be carefully considered such as, effective refractive index inside the channel, propagation length,

and excitation polarization. In the previous study, the Au nanogap geometry was designed only for inducing nanoscale strain gradient on MoS₂ ML. In addition, the excitation polarization was highly perpendicular to the substrate to effectively activate the gap mode plasmon for tip-enhanced photoluminescence spectroscopy (Fig. 1Ra), significantly limiting the SPP coupling in the nanogap (Fig. 1Rb).

Fig. R1. (a) Electric field distribution of tip-metal geometry under excitation polarization parallel to tip axis. Electric field is largely enhanced in z-axis. (b) Electric field distribution of nanogap geometry under same excitation polarization as (a), which cannot effectively launch SPP mode.

4) Figure S2 (supplementary information) couldn't directly justify the following statement "decrease the electron density at the nanogap." For that author should do a similar measurement at three different conditions- air (ambient), vacuum, and oxygen environment and then compare the results for claiming the decrease in electron density in the presence of an oxygen atmosphere.

We thank the reviewer for the helpful comment. Tongay et al. [*Nano Letters* **13**, 2831 (2013)] stated that the high level of annealing is the prerequisite to observe the surrounding gas dependent PL spectra. However, we would not want to try annealing our sample because the required level of annealing can cause the condition change at the interface between the TMD crystal and the the metal substrate which can induce PL quenching effect [*Nanoscale* **12**, 13460 (2020)]. Alternatively, we carefully drop the H₂O onto our sample and measure PL spectra, to confirm the role of surrounding H₂O molecules in decreasing the electron density of our sample. As a result, the dramatic decrease of the trion density with H₂O drop is observed. We have added this discussion in the Supplementary Information with the corresponding description in the main text.

[Revised text] However, at ambient conditions, the electron density of the MoS₂ ML noticeably decreases owing to the presence of H₂O and O₂ molecules physisorbed onto the MoS₂ ML surface [*Physical Review B* **99**, 121201 (2019), *Nano Letters* **13**, 2831 (2013)] (see Fig. S2 in Supplementary Information).

[Added text and Figure in SI]

Fig. S2. PL spectra of MoS₂ ML under ambient condition without (black) and with H₂O molecules (blue) on crystal surface. Zoomed in plot indicate Raman spectra of H₂O molecule.

The PL intensity (trion increasing more faster than exciton) always increases with laser power. Even if you decrease the power to < 0.1 mW, the defect states generate a large number of defect electrons near the Fermi level, which will be capable of forming trions (In the present case, power is very large > 0.1 mW). Did you measure the conversion efficiency (exciton-to-trion) at a low power range ($1 \mu\text{W}$ to $20 \mu\text{W}$) because, at these low powers, the defect electrons are less active, and enhancement of trions may occur entirely due to plasmons?

As discussed in the second query, we have performed additional experiment at the low power ($2 \mu\text{W}$ - $40 \mu\text{W}$) to demonstrate the SPP effect on increasing trion density, while minimizing the role of defect electrons.

[Added results in SI]

Fig. S8. Power dependence of exciton-to-trion conversion. (a) Excitation-power-dependent PL spectra at Au substrate. (b) Extracted X₀ intensity (blue) and X⁻ intensity (red). (c) Extracted

X- intensity at low excitation powers. (d) Excitation-power-dependent PL spectra at SPP mode. (e) Extracted X_0 intensity (blue) and X- intensity (red). (f) Extracted X- intensity at low excitation powers.

To confirm the role of SPP mode on the exciton-to-trion conversion, while excluding the effect of defect-related provision of electrons, we measure excitation-power-dependent PL spectra at Au substrate and SPP mode. At Au substrate, the linewidth of the PL spectra increases as increasing the excitation power, attributed to the emerging trion peak, as shown in Fig. S8a. In contrast to gradually increased X_0 intensity as a function of excitation power, the X- intensity shows negligible changes at the low excitation power ($<10 \mu\text{W}$), which is possibly due to the inactivation of defect-induced electrons at low excitation power (Fig. S8b-c) [*Nano Letters* **19**, 6299 (2019)]. On the other hand, the dominant X- peak are continuously observed with increasing excitation power at the SPP mode, as shown in Fig. S8d. Correspondingly, the X- intensity linearly increases as increasing excitation power even at the low excitation power, as shown in Fig. S8e-f. This behavior well indicates the plasmon-induced hot electron injection [*J. Phys. Chem. Lett.* **8**, 4925 (2017)] and consequently increased X- intensity with high exciton-to-trion conversion efficiency [*Nature photonics* **14**, 324 (2020)].

The authors find another way to justify the statement of "decreased electron density in a nanogap." Did the authors measure the PL in a vacuum? How many times transferred ML MoS₂ on the MIM substrate repeat the measurement? How much gap in an SPP waveguide and Au nanogap in a previous paper (Sci. Adv. 8, eabm5236 (2022)).

We alternatively performed experiment using the H₂O drop on the sample to justify the decreased electron density as discussed earlier. The large area MoS₂ ML is transferred onto the three MIM substrates to confirm the reproducibility of the measurement. The gap size of SPP waveguide in this experiment is $\sim 300 \text{ nm}$, while Au nanogap in the previous paper is $\sim 200 \text{ nm}$.

5) In Figure 3 d, the authors only chose two peaks for fitted the PL spectrum at different polarization. Are there any defects and bi-excitons peaks that have been observed? I believe that at higher power ($> 0.1 \text{ mW}$), defects and bi-excitons (might be) peaks are usually obtained because CVD-grown films always present defects states, and here you transferred CVD ML to another substrate but not getting any defects peaks in any of the power (strange). The authors must in clarifying this issue. What laser power was kept during the measurement of figure 3d data? Was the recorded PL spectrum checked in a vacuum?

We thank the reviewer for pointing this out. We carefully checked the PL spectra in Fig. 3d and could not find the emission from the biexcitons. Specifically, the second derivative of the Fig. 3d (excitation polarization of 45 degree) clearly indicates the existence of two peaks, which can be assigned to the neutral exciton and the trion as reported in the previous study with the similar geometry [*Nature Photonics* **14**, 324 (2020)].

The excitation power was stably kept at $\sim 0.5 \text{ mW}$ during the measurement of Fig. 3d. We guess the defect-related excitons are nonradiatively quenched at room temperature [*Nanoscale* **12**, 3019 (2020)]. We have added this discussion with the second derivative curve in the Supplementary Information.

[Revised text] To further investigate the polarization-dependent behavior of X_0 and X-, we

fit the PL spectra measured at the center of the SPP mode to the Lorentz function, as shown in Fig. 3d (Fig. S9 in Supplementary Information).

[Added result in SI]

Fig. S9. Second derivative curve (top) obtained from PL spectrum in Fig. 3d, with excitation polarization of 45° (bottom). Two minima in second derivative are assigned to neutral exciton peak and trion peak.

6) Did the authors measure the degree of polarization (DOP) of both samples (with and without transfer to MIM)? If not, I suggest doing this for more understanding of the following polarization-dependent exciton-to-trion enhancement phenomena.

We thank the reviewer for suggesting the helpful comment. We measure the degree of polarization of neutral excitons and trions with and without MIM waveguide. Specifically, neutral excitons at SPP mode of MIM waveguide exhibit noticeable polarization degree owing to the SPP-induced excitation of additional X_0 [*Nano Letters* **9**, 3896 (2009), *Optica* **1**, 149 (2014), *Nano Letters* **15**, 5477 (2015)].

[Added text] Finally, with the waveguide fully activated, a highly dominant X- emission is produced, indicating that high-purity localized X- is achieved via a complete exciton-to-trion conversion (Fig. S10-11 in Supplementary Information).

[Added result in SI]

Fig. S10. Polarization degree of excitons and trions. Polar plot for intensity of X_0 (a) and X^- (b) as function of detection angle at Au substrate. Polar plot for intensity of X_0 (c) and X^- (d) as function of detection angle near SPP mode.

To investigate the strain-induced optical characteristics of X_0 and X^- emission, we obtain PL intensities of X_0 and X^- with changing detection angle. At Au substrate, both X_0 and X^- emissions exhibit negligible polarization degree (Fig. S10a-b), in good agreement with previous study [11]. By contrast, at the vicinity of SPP mode, X_0 emission shows the noticeable polarization degree of ~ 0.52 (Fig. S10c) in contrast to still negligible polarization degree of X^- emission (Fig. S10d), attributed to the SPP-induced excitation of additional X_0 [12, 13, 14]. Note that SPP mode is excited with the vertical excitation polarization, which corresponds to waveguide axis.

In Figure 3d, the FWHM of the PL peaks varied a lot from 0° to 90° , justifying the FWHM changes. At 0° polarizing angle, the FWHM is too broad compared with 45° . Why? I suggest authors further check the PL spectra fitting parameters and consider the defects or bi-excitons (or others justify peaks, if available) in a given PL spectra.

We thank the reviewer for pointing this out. We have thoroughly checked the fitting parameters and their error rate. We decrease the degree of freedom to make a reasonable fit for all fitted PL spectra. With regard to the possible existence of other peaks, we could exclude that possibility by confirming the second derivative curve, as discussed in query #5.

[Revised Figure]

Fig. 3. Complete exciton-to-trion conversion. (a) Illustration of exciton-to-trion conversion process assisted by plasmon-induced hot electrons. (b) Normalized TRPL traces of MoS₂ monolayer on silicon (black) and waveguide structure (red). (c) Spatial dependent PL spectra obtained by vertically crossing waveguide (white dashed line) with different excitation polarizations. (d) Corresponding PL spectra obtained at center of SPP mode fitted to Lorentz function.

Fig. 4. All-optical control of exciton-trion interconversion. (a) Illustration of spatio-spectral modulation of SPP mode and excitonic emission response through adaptive wavefront shaping. (b) Evolution of SPP intensity during stepwise sequence feedback. Inset: optimized phase mask (PM) after sequence feedback (see Fig. S5 in Supplementary Information). (c) SPP spectra before (black) and after (red) wavefront shaping. (d) Corresponding PL spectra before (left) and after (right) wavefront shaping fitted to Voigt function. Black, blue, and red lines indicate fit of raw spectrum, X₀, and X⁻, respectively. Black dots indicate raw data. (e) Time-series normalized PL spectra during on/off switching of optimal phase mask obtained in (b). (f) Corresponding X⁻/X₀ ratio.

7) The objective of the present manuscript is to enhance the trion signal/density (by reducing exciton) in a PL spectrum, so the most essential question, enhancement of trions, are singlet or triplet? I suggest that the authors should discuss a few lines in the text about the type of trions. How are the hot electrons affecting the transition at K and K'? Do authors have any phenomenal model for explaining the hot electron (or SPP mode) coupling with excitons (for forming trions) at K and K' (interband and intraband transition roles)?

Our MIM waveguide geometry activates SPP mode with linearly polarized excitation at room temperature, resulting in the superposition of the two valley states and therefore mixed states of singlet and triplet trions [*npj 2D Materials and Applications* **6**, 27 (2022), *Nature Nanotechnology* **8**, 634 (2013)]. We have found a few studies demonstrating valley-selective hot electron injection, which in turn facilitating selective generation between the singlet and triplet trions [*Physical Review B* **105**, 075434 (2022)].

However, these approaches are based on circularly polarized excitation, which is currently unavailable in our geometry. MIM waveguide in our experiment is highly sensitive to excitation polarization, as indicated in Fig. 2 and 3. Circularly polarized excitation cannot guarantee the activation of SPP mode in our MIM waveguide. We are currently in progress of designing new MIM waveguide as a future work, which can facilitate the activation of SPP mode with circularly polarized excitation and consequently lead to the generation of valley-selective trions [*Nature Nanotechnology* **17**, 1178 (2022)].

Phonon plays an important role in the formation of trions. Is there any exciton-phonon interaction occurring for the coupling of SPP electrons for the making of trions (I mean, the picture is not so simple that all-time excitons couple with SPP electrons and form trions)? Could any Fermi level affecting while interacting with the plasmon signal? If yes, explain.

With regard to the role of exciton-phonon interaction for the trion formation, Niehues et al. [*Nano Lett.* **18**, 1751 (2018)] reported that applying strain with the range of 0.33-1.15 % and corresponding modification of exciton-phonon coupling gives a negligible effect on the trion formation (Fig. S5 in *Nano Lett.* **18**, 1751 (2018)). Therefore, in our geometry with the applied strain of ~0.1 %, we can exclude the effect of exciton-phonon interaction.

In our geometry, we demonstrate the increased exciton-to-trion conversion ratio at the nanogap center through the strain-induced modification of the bandgap, which confines excitons and electrons toward the nanogap center. It can increase their spatial overlap resulting in the increased trion generation [*Nature Photonics* **14**, 324 (2020)].

However, as the reviewer pointed out, the strain also induces change in the band structure and accordingly affect the metal-semiconductor interaction, leading to change in the efficiency of plasmon-induced hot electron injection. In our experiment, the tensile strain in the geometry reduces Schottky barrier height [*Physical Chemistry Chemical Physics* **17**, 27088 (2015)], which affects to Fermi level. Plasmon can also indirectly affect Fermi level as they increase the electron density, which pushes Fermi level toward conduction band [*Journal of Materials Chemistry C* **9**, 11407 (2021)]. We have added the detailed description for the mechanism of enhanced trion density in the main text.

[Revised text] The efficiency of this exciton-to-trion conversion can reach 100 % in a WS₂ ML suspended on a microhole-based strain gradient, **because the strain-induced modification of bandgap increases the spatial overlap between X₀ and electrons [*Nature Photonics* **14**, 324 (2020)] and affects Fermi level in the way of decreasing Schottky barrier height [*Physical***

Chemistry Chemical Physics **17**, 27088 (2015)].

8) Has the phase mask been implemented via SLM incorporated with light? The SPP intensity has been measured along with PL intensity (setup shown in Fig. 1 a). If yes, then how to distinguish the signal because both the emissions (ML PL and SPP) are in the same range?

We apologize for the lack of description on obtaining optimal phase mask. We fabricated identical MIM substrates, as shown in Fig. S12. The optical phase mask for increasing SPP intensity was obtained at the sample #1 without the MoS₂ ML. Then, we move onto the sample #2 with MoS₂ ML and measure PL spectra with optical phase mask at the exactly the same position as the sample #1. We have added this discussion with detailed optical microscope image of our sample in the Supplementary Information to deliver clear information to readers.

[Revised text] We move the detection spot to the weak SPP region and implement a sequence feedback algorithm, which aims to maximize the target intensity by optimizing the wavefront (Fig. S12-13 in Supplementary Information) [*Nature Communications* **12**, 3465 (2021), *Advanced Materials* **33**, 2008234 (2021)].

[Added results in SI]

Fig. S12. Optical microscope image of lateral MIM waveguide device. MoS₂ ML is transferred on the waveguides #2-4 excepting the waveguide #1.

We fabricate four identical lateral MIM waveguide structure (sample #1-4). The MoS₂ ML is transferred on the three lateral MIM waveguides (sample #2-4) while sample #1 remains bare. As shown in Fig. 4c-d, SPP signal is spectrally overlapped with MoS₂ PL. Therefore, to find the optical phase mask with the SPP signal without the influence of MoS₂ PL, we used the waveguide #1. Because all lateral MIM waveguides are identical, they have identical SPP mode and share an optimal phase mask. Therefore, we optimize the phase mask at the device #1 and then move onto the device #2 to perform the main experiments.

9) The ML WS₂ on to Si/SiO₂ substrate at low temperature (<77 K) becomes an excitonic rich in nature (means: more excitons and less or no trions), so what authors believe that the proposed conversion scheme would work in this case? The author did observe a similar MIM

scheme for other similar MX₂/WX₂ types of materials (for a generalized proposed scheme robust).

The recent study by Harats et al. [*Nature Photonics* **14**, 324 (2020)] demonstrated efficient exciton-to-trion conversion through microscale strain-gradient on SiN/Si substrate. Because MIM substrate in our experiment uses similar strain-gradient geometry with additional functionality to provide extra electrons, we believe that our work provides a generalizable approach.

10) In Fig S10 a, representing $n(x)$ is a spatial distribution of the exciton with (red) and without (black) strain gradient, while in the text, authors are writing $n(x)$ is the summation of X₀ and X⁻ spatial density distribution i.e., $n(x) = n_{ex}(x) + n_{tr}(x)$. Clarify the notification and statement.

We apologize for causing a confusion to readers. $n(x)$ is the solution of the drift-diffusion model, indicating the density of photoexcited excitons. Photoexcited excitons includes the neutral excitons and charged excitons. To investigate their individual density, the mass action model should be further considered. In Fig. S18a (Fig. S10 has been modified to Fig. S18), the mass action model is not considered yet. Therefore, it indicates the spatial distribution of the all photoexcited excitons, which confirming the role of the strain-gradient by the nanogap. Then, Fig. S18b-c now includes the mass action model to show the ratio of neutral exciton (X₀) and charged exciton (X⁻). We acknowledge that the use a terminology “exciton density” to represent the $n(x)$ can cause the misunderstanding to readers. Therefore, we have thoroughly reviewed the text and substitute it to “total photoexcited excitons” with detailed explanation.

[Added text] We note that, at this stage, the photoexcited exciton density $n(x)$ includes all kinds of photoexcited excitons, e.g., neutral excitons (X₀) and charged excitons (X⁻).

[Revised text] In this case, we now consider the contribution of X⁻ because the actual solution of drift-diffusion model $n(x)$ is the summation of X₀ and X⁻ densities, i.e., $n(x) = n_{ex}(x) + n_{tr}(x)$.

[Revised text in SI]

Fig. S18. Estimation of spatial distribution of X₀ and X⁻. (a) Spatial distribution of the photoexcited excitons before considering mass action model with (red) and without (black) strain gradient. Contribution of X₀ (green) and X⁻ (red) in total photoexcited exciton density (black) for $\alpha = 0.04$ (b) and $\alpha = 1.2$ (c).

As mentioned in the main text, we subtract the photoexcited exciton density obtained without the strain gradient (black, in Fig. S18a) from the exciton density with the strain gradient (red, in Fig. S18a) to exclude the effect of the optical excitation. At this stage, we exclude the contribution of X⁻ to clearly investigate the role of strain gradient by the nanogap. Then, we adopt the mass action model to estimate the density ratio of X₀ and X⁻ depending on the background electron density α , as shown in Fig. S18b and S18c.

11) In the theoretical section, so many notations are not well appropriately described, such as ϕ (work function usually defined as $E_{\text{vac}} - E_F$) is the same here or different definition has been considered by authors. What are n_A and n_B , and how is it affecting the experimental data? The authors have experimentally estimated n_B (in Fig 5 e), but the objective and correlation are not clear well.

We thank the reviewer for pointing out our mistakes. Our intention of using ϕ was to define work function. To clearly deliver this information, we have added the additional description about the work function in the revised manuscript as follows:

[Added text] Note that work function $\phi = E_{\text{vac}} - E_F$, where E_{vac} is the vacuum level and E_F is the Fermi level.

With regard to n_B , it was the misdefinition and should be substitute to n_e as we defined in equation (2). n_e is the density of electron and directly related to the density of trion by equation (3). Increasing n_e indicates the enhanced trion density and this relation is clearly demonstrated in the previous study [Fig. S3 in *Nature Communications* **4**, 1474 (2013)]. n_A is the relation from the law of mass action model connecting the quantities of neutral and charged excitons and free electrons [*Physical Review* **B** 59, 1602 (1999)]. This is predefined by the trion binding energy and temperature. Decreasing temperature or increasing trion binding energy possibly lead to the decrease in n_A , consequently resulting in decreased trion density by the equation (3).

12) Fig 5 a, b, and c have been estimated via KPFM, and the meaning of ϕ is the same as I defined above. If yes, so the max. value of the work function is around 5.4 in a gap (suspended ML MoS₂) but the estimated value (by others) is 6.11 eV, in slight the difference of the work functions and briefly explains the details of measurement of KPFM for more clearance of reader.

The work function of MoS₂ ML has a variation with many parameters such as surrounding conditions [*ACS Nano* **10**, 6100 (2016)] and the type of substrates [*Nanotechnology* **30**, 245708 (2019)]. For the MoS₂ ML on SiO₂ substrate, it revealed to have a range of ~4.8-5.2 eV, similar to our observation. Besides, Kwon et al. [*Scientific Reports* **9**, 14434 (2019)] reported a decrease in work function of MoS₂ ML on the Au substrate due to the interfacial electric dipole energy. We acknowledge the lack of description on the KPFM measurement. Consequently, we have added the detailed description of KPFM measurement in the Methods as follow:

[Added text] We measured the work function of the region of a gap (suspended ML MoS₂) sample by using a conventional AFM (Park Systems Co., NX10) with a measurement mode of Kelvin probe force microscope (KPFM) addressing nanoscale surface work function. We used a commercial Cr & Au coated cantilever (Mikromasch Co., NSC18). Before the main measurement, we performed a calibration on the work function of cantilever by scanning the highly ordered pyrolytic graphite (HOPG, ~4.6 eV), commonly used as calibration method for KPFM. Then, we performed the nanoscale work function mapping on the suspended ML MoS₂ with the calibrated cantilever showing maximum value of about 5.4 eV. Note that the KPFM measurement was performed without optical excitation to precisely characterize the work function of MoS₂ ML without external perturbation.

During the measurement of KPFM, applying the voltage near the tip may tune the Fermi level of the material and also may create local defects, resulting producing unbound electrons that will easy to couple with excitons. May it could be the reason for enhancing the trions more coupled with the SPP plasmons? What is the vision of the authors regarding the above statement?

KPFM measurement was performed independently from other optical measurements. Therefore, we exclude the possibility from unexpected creation of local defects by KPFM while obtaining PL spectra in the manuscript. In order to prevent the confusion to readers, we have added the detailed description on the KPFM measurement in the Methods section.

Is there any correlation between Fig 5 e (main text) and Fig S10 (supporting document)? Has the difference of Fig S10 data presented in Fig 5 e? I suggest the authors should rewrite the theoretical part properly (explain all the parameters properly) and make a clear bridge between theoretical and experimental results.

We apologize for raising such concerns. We have thoroughly revised the theoretical part and added descriptions to clearly demonstrate all parameters. With regard to the specific concern about Fig. 5e and Fig. S18 (previously Fig. S10), Fig. 5e (black) indicates the density of total photoexcited excitons obtained by subtracting the density of total photoexcited excitons without strain gradient from with strain gradient, as shown in Fig. S18a. This is for clearly showing the drifted photoexcited excitons by strain gradient, while excluding the effect of optical excitation.

[Revised text] Note that we subtract the photoexcited exciton density obtained without the strain gradient from the photoexcited exciton density with the strain gradient to clearly demonstrate the effect of strain gradient and consequently evaluate the drifted photoexcited excitons while excluding the effect of optical excitation (Fig. S18 in Supplementary Information).

Reviewer #2:

This manuscript demonstrated an all-optical control method for the exciton-to-trion conversion and spatial distribution control in a MoS₂ monolayer. By designing nanoscale strain gradient in MoS₂ monolayer transferred on a metal–insulator–metal (MIM) waveguide, they accomplished the efficient electron density and exciton-to-trion conversion. In addition, by using adaptive wavefront shaping, the surface plasmon polaritons (SPPs) mode can be modulated, so that an all-optical control of the exciton-to-trion conversion rate and trion distribution can be realized. The all-optical method for excitonic quasiparticle control proposed by this work can be quite useful for excitonic photoconversion. The results are original and new in the relevant research community, and have significance in terms of potential nanophotonics application. I would recommend further consideration of this manuscript for publication in Nature Communication, after authors address the following questions/comments:

We thank the reviewer for acknowledging the novelty and significance of our work. We also appreciate the constructive comments to improve the completeness of our manuscript. With regard to the raised concerns, we have addressed each of the comments and made corresponding revisions to our manuscript.

1) Since there have been quite a few reports for the electric gate-controlled exciton/trion PL efficiency in terms of TMD ML materials, a general question is about the exciton-to-trion conversion rate comparison, between the electric control method such as application of an electric gate, chemical doping method, and the all-optical control method demonstrated in this work. It is better to give some comments/comparison for the electric, chemical and all-optical control methods for the trions generation.

We thank the reviewer for the helpful comment. The efficiency of exciton-to-trion conversion can be highly affected by the spatial overlap between the neutral excitons and electrons. Harats et al. [*Nature Photonics* **14**, 324 (2020)] revealed that the efficiency of the exciton-to-trion conversion can reach ~100 % with strain gradient geometry. Therefore, in contrast to the electrical gating [*ACS Nano* **9**, 647 (2015)] and chemical doping [*Nano Letters* **13**, 5944 (2013)] methods, our all-optical control modality with strain gradient can have higher exciton-to-trion conversion efficiency, requiring a lower level of electron doping. We have added this discussion in the revised main text to emphasize the advantages of our all-optical control method compared to other electrical and chemical method as follows:

[Added text] In comparison with the large-area electrical [*ACS Nano* **9**, 647 (2015)] and chemical doping methods [*Nano Letters* **13**, 5944 (2013)], exploiting strain gradient geometry facilitates higher conversion efficiency, local injection of the electrons, and efficient spatial controllability, suitable to applications in nanoscale optoelectronic devices and trionic energy harvesting.

2) Since the plasmonic MIM waveguide is utilized in this work to enable the plasmon-induced hot electron generation and injection to the MoS₂, and therefore the additional exciton-to-trion conversion. Could authors give some rough estimation about the generated and injected electron density? And correspondingly, the exciton-to-trion conversion efficiency? Though the calculated exciton and electron densities are shown in Fig. 5e & 5g, there are no given absolute values.

We thank the reviewer for pointing this out. As Harats et al. [*Nature Photonics* **14**, 324 (2020)] reported earlier, the exciton-to-trion conversion efficiency can reach $\sim 100\%$ at strain gradient. This can be seen in Fig. 5g, showing the increased trion density with the decreased neutral exciton density as electrons increase. It indicates that, the number of injected electrons should roughly follow the number of newly generated trions. Therefore, we estimate the number of injected electrons at Fig. 5h (red dashed line) based on the experimentally obtained $X-/X_0$ ratio. We have added this discussion with a updated figure in the revised main text.

[Revised Figure]

Fig. 5. Theoretical investigation of electron funneling and exciton-to-trion conversion dynamics. Topography (a) and work function (ϕ) images (b) of waveguide obtained by KPFM. (c) Work function profile derived from dashed green line in (b). (d) Height profile of MoS₂ ML on nanogap of waveguide (green dots), fitted line shape function (black line), and height profile of nanogap without MoS₂ ML (yellow line). (e) Spatial density distribution of photoexcited excitons (black) and electrons (red) under strain profile estimated from fitted line-shape function in (d). (f) Spatial electron density distribution as function of global defect density α for estimated strain profile. (g) X₀ density (blue) and X⁻ (red) density as functions of α . (h) X⁻/X₀ ratio as function of α . Dashed black line represents theoretically obtained fit from (g). **Dashed red line indicates theoretically driven electron density at center of nanogap, i.e., $n_e(0)$ as function of α .** Orange triangles and navy squares indicate experimentally obtained values from Fig. 3 and 4, respectively. Inset: closed view of blue-filled region; values at bottom left are without optimal phase mask, whereas those at top right are with optimal phase mask.

[Added text] By assuming $\sim 100\%$ exciton-to-trion conversion efficiency under the strain gradient [*Nature Photonics* **14**, 324 (2020)], we quantitatively estimate the electron density based on the experimentally obtained $X-/X_0$ ratio, exhibiting a maximum 10-fold enhancement of the localized electron density [*Science Advances* **5**, eaax9958 (2019)].

3) The Fig. S1 in the Supplementary Information, shows the Raman spectra of MoS₂ ML transferred on Au substrate and suspended on trionic waveguide. It would be good to have the optical reflectivity spectra for MoS₂ ML transferred on Au substrate and suspended on trionic waveguide as well. With the reflectivity and/or PL spectra comparison, could authors give some discussions/comments about the possible energy transfer between the SPP and

excitons/trions of MoS2 ML? And how this possibly may affect the exciton-to-trion conversion?

We thank the reviewer for suggesting interesting experiment. There have been studies, demonstrating the energy transfer between SPP and excitons [*Optica* **1**, 149 (2014), *Nano Letters* **15**, 5477 (2015)]. In these works, the energy transfer between the SPP and the excitons was demonstrated by showing the PL enhancement at the SPP region. Similarly, we obtained the PL spectra with and without the SPP mode and investigated the changes in neutral exciton and trion density. We have found the additionally generated trions, which can be attributed to the SPP-induced generation of excitons and their consequent conversion to trions. We have included this discussion in the revised Supplementary Information with a brief description in the revised main text.

[Added text] We note that the increase in X- intensity is higher than the decrease in X₀ intensity with the optimal phase mask. This is due to the SPP-induced excitation of additional X₀, consequently converted to X- (Fig. S10-11 in Supplementary Information).

[Added results in SI]

Fig. S11. Change in trion density (ΔX^-) as function of change in neutral exciton density (ΔX_0), with activation of SPP mode.

To investigate the possible energy transfer between the excitons and the SPP, we obtain the PL spectra with and without the SPP mode. We then extract X₀ and X- intensities by fitting PL spectra with Lorentz function. Fig. S11 shows the change of trion density (ΔX^-) as a function of the change of neutral exciton density (ΔX_0). Without the influence of the SPP, decrease in X₀ density should correspond to increase in X- density with dominating highly efficient exciton-to-trion conversion process at strain gradient geometry. However, Fig. S11 exhibits the extra increase in the trion density. This is probably due to the SPP-induced excitation of excitons [*Optica* **1**, 149 (2014), *Nano Letters* **15**, 5477 (2015)] and these additionally generated excitons are converted to trions as enough number of electrons are confined at the center of nanogap.

4) With the plasmonic MIM waveguide design, the funneled electrons and converted trions, most likely, are localized (in the nanogap center). Even the high exciton-to-trion conversion rate can be obtained by this method, the converted trions would have even lower diffusion length, it hardly expects any exciton/trions flux that can be useful for promising exciton-based optoelectronic devices when exciton flux is concerned. The authors may consider this issue and have some corresponding statement in their Introduction part, and the Discussion part as

well.

We thank the reviewer for sharing the insightful perspective. As pointing out, generating flux of the converted trions will facilitate further applications in exciton/trion flux-based optoelectronic devices and circuits. Therefore, as a follow-up work, we are currently in progress of manipulating converted trions by conducting the sharp tip of atomic force microscopy (AFM). We have preliminary simulation data, demonstrating the concept of controlling spatial distribution of the converted trions. Here, the conductive tip is located at the left side of the nanogap. When the positive bias is applied to the tip, the converted trions across the nanogap can move toward the left direction, i.e., attracted to the tip. The movement of the converted trions can be further modulated by applying opposite direction of the bias on the tip or changing the tip position.

Fig. R2. (a) General schematic of tip-induced trion control platform. (b) Electric field distribution with applying positive bias on tip. (c) Vector map of electric field.

5) The exciton-to-trion conversion rate and trion distribution control realized by the SPPs mode modulation demonstrated in this work is excellent result. This result suggests potential nanophotonics application for TMD materials integrated with a plasmonic MIM waveguide. This might be more promising than that of exciton/trions flux-based optoelectronic devices, as mentioned in the previous comment. So I would suggest the authors comment this issue by referring the other relevant works for nano structures integrated with a plasmonic MIM waveguide.

We again sincerely thank the reviewer for the positive evaluation on our work. Based upon this constructive comment, we have included the perspective for the plasmonic MIM waveguide applications as potential alternatives of existing exciton/trion flux-based optoelectronic devices in the Discussion section of the revised manuscript.

[Added text] Meanwhile, generating trionic flux with the converted trion should be a pressing matter, as it opens a pathway toward manipulating excitonic/trionic flux efficiently at the nanoscale combined with existing plasmonic MIM waveguide geometry [*Nature* **440**, 508 (2006), *Nano Letters* **19**, 7632 (2019), *Light: Science & Applications* **4**, e294 (2015)].

Reviewer #3:

The manuscript entitled " All-optical control of high-purity trions in nanoscale waveguide" by Lee et al. have investigated the generation of high-purity trions and dynamic exciton-trion interconversion by a designed metal-insulator-metal (MIM) waveguide with spatial controllability of the surface plasmon polariton (SPP) mode facilitated with spatial light modulator (SLM).

In this study, the authors investigated the monolayer MoS₂ suspended on the nanogap geometry of the plasmonic MIM waveguide which induces the strain gradient leading to the significantly increases the funneling efficiency for confining excitons to the nanogap. The concepts and related works have been published by the same group as listed in references 13 and 14. They further utilized the SPP mode of the plasmonic MIM waveguide to induce additional hot electrons to enable the injection of electrons from Au to the MoS₂. It was observed that the extra electrons are funneled toward the nanogap and locally increase the exciton-to-trion conversion. The polarization-dependent experiments are also performed to compare the trion conversion rate with respect to the magnitude of activation SPP modes. Moreover, the authors employ a SLM with adaptive wavefront shaping and sequence feedback algorithm to spatially control the trion emission intensity via locally enhance the plasmon intensity in the weak SPP region. The physical mechanism of electron funneling and the related exciton-to-trion conversion dynamics is analyzed by drift-diffusion model and experimentally obtained Kelvin probe force microscopy (KPFM) data. Theoretical estimation of the spatial distribution of the electron and X₀ in the presence of the nanoscale strain gradient supports experimental data.

The manuscript is well-written and well-organized with the step by step realizing the all optical control high-purity of tions dynamically and locally. Their scientific aim is precise. The work is significant to the field of optical control low-dimensional nanodevice applications. It is a pioneer work considering the adaptive wavefront shaping technique into optoelectronic materials. The work exploits SPP mode of the plasmonic MIM waveguide and 1D nanoscale strain gradient in suspended MoS₂ ML. It shows a better control of the ratio of dominant X₀ emission and high-purity X- emission compared to other published results elsewhere important for exciton-based optoelectronic devices. The work supports the conclusions and claims. The methodology looks sound and matches with the standard of our field. The analysis is good and the theoretical estimations support the observed results. However, there are some comments/issues that need to be addressed before accepting for publication. The work meets the expected standards for the journal of Nature Communications if the mentioned concerns can be carefully addressed.

We thank the reviewer for the positive evaluation on our work with high praise. We also appreciate the constructive comment to improve the completeness of our manuscript. With regard to the raised concerns, all authors have carefully discussed and performed additional experiments to make corresponding revisions on our manuscript and Supplementary Information.

1) In the method description, the low power CW He-Ne laser (594.5 nm, <1.0 mW) was used to excite the sample. How to use a low power He-Ne laser instead of tip to activate the SPP mode? Is this the reason why have the weak SPP region? The SPP formation by laser needs to be further classified and compare with tip induced SPP mode.

We thank the reviewer for raising this important question. In the laser illumination, there are

two mechanisms responsible for light coupling to the nanogap and exciting the sample. First, the nanogap MIM geometry (Au-SiO₂-Au) has been designed to provide a high contrast between the effective refractive index inside the channel and the refractive index of the substrate ($n_{\text{SiO}_2} = 1.46$), providing a high efficiency coupling. Furthermore, coupling of incoming laser light with the MIM nanogap to activate the SPP mode proceeds through scattering from the edges and sidewalls of the gap for effective momentum matching. This method, where light is coupled to Au-SiO₂-Au MIM nanogap through edge / tail-end illumination, has been discussed in detail in previous works including:

Choo, H. et al. "Nanofocusing in a metal–insulator–metal gap plasmon waveguide with a three-dimensional linear taper." *Nature Photonics* **6**, 838 (2012).

Bao, W. et al. "Mapping local charge recombination heterogeneity by multidimensional nanospectroscopic imaging." *Science* **338**, 1317 (2012).

Kumar, S. et al. "Overcoming evanescent field decay using 3D-tapered nanocavities for on-chip targeted molecular analysis." *Nature communications* **11**, 2930 (2020).

To improve these coupling conditions and minimize losses in SPP propagation, the width and height of the gap were carefully optimized in our simulations and fabrications of the device.

[Revised text] By contrast, the proposed nanogap-based lateral MIM waveguide device with the designed SPP mode can supply extra electrons locally via plasmon-induced hot electron generation, as illustrated in Fig. 1b (see Fig. S3 in Supplementary Information).

[Added results in SI]

Fig. S3. Fabrication process of lateral MIM waveguide.

The coupling of light and SPP propagation rely on a lateral MIM geometry where the top and sidewalls of the nanogap are gold-coated, whereas bottom of the channel is SiO₂. For fabrication of the channel, we start with a SOI wafer, and deposit 150 nm of Au on it. Then we perform a FIB milling step to etch into the SiO₂. As seen in the Fig. S3, after milling the resultant nanogaps have a thin layer of gold on top, but much of the sidewall is SiO₂. This milling process results in a slight taper to the sidewalls, which is crucial for subsequent gold deposition step. The top layer of gold is then removed using a gold etchant. A fresh layer of Au (50 nm) is deposited using e-beam, and coats the top, sidewalls, and bottom of the nanogap. A second round of milling is then performed to remove the gold from bottom of the nanogap channel. As the figure illustrates, the first milling process is an oxide etch step, whereas the second milling is an Au etch step to remove gold from bottom of the channels.

The parameters for fabricating lateral MIM waveguide, such as gap size and height, are optimized in the way of maximizing the efficiency of SPP coupling to the nanogap while also

minimizing losses during SPP propagation in the nanogap. Specifically, the nanogap lateral MIM geometry (Au-SiO₂-Au) has been designed to provide a high contrast between the effective refractive index inside the channel and the refractive index of the substrate (nSiO₂ = 1.46) providing a higher coupling efficiency. Furthermore, coupling of incoming laser light with lateral MIM nanogap to activate SPP mode proceeds through scattering from the edges and sidewalls of the gap. This method where light is coupled to Au-SiO₂-Au MIM nanogap through edge/tail-end illumination has been discussed in detail in previous works [*Nature Photonics* **6**, 838 (2012), *Science* **338**, 1317 (2012), *Nature communications* **11**, 2930 (2020)].

2) The work focuses on the all optical control method however the optical power dependence is not considered in the experiment data. I suggest the authors perform the laser power dependence experiments to further conclude their findings if the hot electrons are highly dominant in the nanogap center.

We thank the reviewer for the constructive comment. The electron density can also be increased by various origins, such as defects and surrounding conditions, and it will be important to exclude these possibilities. Therefore, we obtain excitation power dependent PL spectra at two locations, i.e., Au substrate and SPP mode. At the low excitation power (<10 μW), defect states are likely to be inactive [*Nano Letters* **19**, 6299 (2019)], exhibiting negligible trion emission without the contribution of SPP. By contrast, the trion emission continuously increases at SPP mode even at low excitation power, confirming the role of hot electrons in our experiment. We have added this discussion in the Supplementary Information with corresponding description in the revised main text.

[Added text] Note that we exclude the possible contribution from defect-induced charges while confirming the role of lateral MIM waveguide with control experiments at low excitation power (see Fig. S8 in Supplementary Information).

[Added results in SI]

Fig. S8. Power dependence of exciton-to-trion conversion. (a) Excitation-power-dependent PL

spectra at Au substrate. (b) Extracted X_0 intensity (blue) and X^- intensity (red). (c) Extracted X^- intensity at low excitation powers. (d) Excitation-power-dependent PL spectra at SPP mode. (e) Extracted X_0 intensity (blue) and X^- intensity (red). (f) Extracted X^- intensity at low excitation powers.

To confirm the role of SPP mode on the exciton-to-trion conversion, while excluding the effect of defect-related provision of electrons, we measure excitation-power-dependent PL spectra at Au substrate and SPP mode. At Au substrate, the linewidth of the PL spectra increases as increasing the excitation power, attributed to the emerging trion peak, as shown in Fig. S8a. In contrast to gradually increased X_0 intensity as a function of excitation power, the X^- intensity shows negligible changes at the low excitation power ($<10 \mu\text{W}$), which is possibly due to the inactivation of defect-induced electrons at low excitation power (Fig. S8b-c) [*Nano Letters* **19**, 6299 (2019)]. On the other hand, the dominant X^- peak are continuously observed with increasing excitation power at the SPP mode, as shown in Fig. S8d. Correspondingly, the X^- intensity linearly increases as increasing excitation power even at the low excitation power, as shown in Fig. S8e-f. This behavior well indicates the plasmon-induced hot electron injection [*J. Phys. Chem. Lett.* **8**, 4925 (2017)] and consequently increased X^- intensity with high exciton-to-trion conversion efficiency [*Nature photonics* **14**, 324 (2020)].

3) A strong SPP mode is observed in the waveguide, as shown in Fig. 2b. How to measure the SPP mode in the experiment? It is helpful to readers if the authors describe the measurement of SPP modes in detail.

We thank the reviewer for pointing this out. We acknowledge the lack of description on obtaining the SPP mode in our experiment, even though it is important process to measure SPP signal in Fig. 2 and obtain optimal phase mask in Fig. 4. We have added the detailed description and procedure of the SPP measurement with the geometry of our sample in the revised Supplementary Information.

[Revised text] We move the detection spot to the weak SPP region and implement a sequence feedback algorithm, which aims to maximize the target intensity by optimizing the wavefront (Fig. S12-13 in Supplementary Information) [*Nature Communications* **12**, 3465 (2021), *Advanced Materials* **33**, 2008234 (2021)].

[Added results in SI]

Fig. S12. Optical microscope image of lateral MIM waveguide device. MoS₂ ML is transferred on the waveguides #2-4 excepting the waveguide #1.

We fabricate four identical lateral MIM waveguide structure (sample #1-4). The MoS₂ ML is transferred on the three lateral MIM waveguides (sample #2-4) while sample #1 remains bare. As shown in Fig. 4c-d, SPP signal is spectrally overlapped with MoS₂ PL. Therefore, to find the optical phase mask with the SPP signal without the influence of MoS₂ PL, we used the waveguide #1. Because all lateral MIM waveguides are identical, they have identical SPP mode and share an optimal phase mask. Therefore, we optimize the phase mask at the device #1 and then move onto the device #2 to perform the main experiments.

4) “Unlike previously reported plasmon-coupled platforms, which demonstrated dramatic decreases in lifetime, both components derived from the waveguide exhibit minimal changes in decay time compared with the ones from silicon. This is attributed to the strain gradient geometry, which can have higher electron densities even without being strongly coupled to the plasmon.”

Although the paragraph explains the TRPL result, it is still not straightforward to connect the relation of higher electron densities and life time of exciton or trion in this description. It should be further clarified for clear understanding.

We thank the reviewer for pointing this out. Here, we would like to emphasize the advantages of the strain gradient geometry, facilitating high electron density without strongly coupled to the plasmon. While both plasmon-coupled platform and our MIM waveguide can induce high electron density, plasmon-coupled platform can cause significant decrease in decay time [*Nature Nanotechnology* **13**, 1137–1142 (2018), *Advanced Optical Materials* **8**, 2001147 (2020)], limiting applications in energy harvesting and flux-based devices/circuits. To clearly deliver our intention to readers, we have carefully revised the statement in the main text as follows:

[Revised text] Unlike previously reported plasmon-coupled platforms, exhibiting significant decreases in decay time [*Nature Nanotechnology* **13**, 1137–1142 (2018), *Advanced Optical Materials* **8**, 2001147 (2020), *ACS Nano* **16**, 14390 (2022)], both components derived from lateral MIM waveguide show minimal changes in decay time compared to the ones from silicon. Specifically, the strain gradient geometry exploits the funneling of electrons and high exciton-

to-trion conversion efficiency [*Nature Photonics* **14**, 324 (2022)], resulting in the smaller number of injected electrons to achieve complete exciton-to-trion conversion. Therefore, we induce high electron density and correspondingly enhanced trion emission while weakly coupled to the plasmon, as shown in Fig. 3b.

5) The descriptions of all the figures in main text should be more elaborative. For example, n_B in Fig. 5e needs to define in the main text properly.

We thank the reviewer for the constructive comment to improve the completeness of our manuscript. We have thoroughly checked the theoretical part and added the detailed descriptions to clearly demonstrate all the parameters. With regard to the specific concern on n_B , it was the typo and should be substitute to n_e as we defined in equation (2).

[Revised figure]

Fig. 5. Theoretical investigation of electron funneling and exciton-to-trion conversion dynamics. Topography (a) and work function (ϕ) images (b) of waveguide obtained by KPFM. (c) Work function profile derived from dashed green line in (b). (d) Height profile of MoS₂ ML on nanogap of waveguide (green dots), fitted line shape function (black line), and height profile of nanogap without MoS₂ ML (yellow line). (e) Spatial density distribution of photoexcited excitons (black) and electrons (red) under strain profile estimated from fitted line-shape function in (d). (f) Spatial electron density distribution as function of global defect density α for estimated strain profile. (g) X₀ density (blue) and X⁻ (red) density as functions of α . (h) X⁻/X₀ ratio as function of α . Dashed black line represents theoretically obtained fit from (g). **Dashed red line indicates the theoretically driven electron density at the center of nanogap, i.e., $n_e(0)$ as a function of α .** Orange triangles and navy squares indicate experimentally obtained values from Fig. 3 and 4, respectively. Inset: closed view of blue-filled region; values at bottom left are without optimal phase mask, whereas those at top right are with optimal phase mask.

[Added text] Note that work function $\phi = E_{\text{vac}} - E_F$, where E_{vac} is the vacuum level and E_F is the Fermi level.

[Added text] We note that, at this stage, the photoexcited exciton density $n(x)$ includes all kinds of photoexcited excitons, e.g., neutral excitons (X₀) and charged excitons (X⁻).

[Revised text] In this case, we now consider the contribution of X- because the actual solution of drift-diffusion model $n(x)$ is the summation of X_0 and X- densities, i.e., $n(x) = n_{ex}(x) + n_{tr}(x)$.

[Revised text] Note that we subtract the photoexcited exciton density obtained without the strain gradient from the photoexcited exciton density with the strain gradient to clearly demonstrate the effect of strain gradient and consequently evaluate the density of drifted photoexcited excitons while excluding the effect of optical excitation (see Fig. S10 in Supplementary Information).

[Revised text] If we define this global defect density of the sample as α , then an increasing α indicates the provision of extra electrons, i.e., α is proportional to the electron density $n_e(x)$.

[Revised text] where $n_A = \frac{4m_{ex}m_e}{\pi\hbar m_{tr}} k_B T e^{-E_T/k_B T}$ (m_{ex} , m_{tr} , and m_e are the masses of X_0 , X-, and electron) is the relation of connecting concentrations of X_0 , X-, and electrons by the law of mass action.

6) The defect density α is related to n_B and plays an important role for generating hot electrons and increase in background electrons. A little bit of more discussions on a) the relation between α vs n_B and b) how one can experimentally manipulate defects in such systems in a controllable way, are suggested to include in the text.

We thank the reviewer for the constructive comment. n_e (previously mis-defined as n_B) is the density of electron and directly related to the density of trion by equation (3). Increasing n_e indicates the enhanced trion density and this relation is clearly demonstrated in the previous study [Fig. S3 in *Nature Communications* **4**, 1474 (2013)]. The way of increasing n_e is increasing defect density α , as they linearly proportional to each other by equation (2). While there have been studies, directly manipulating the defect density in MoS_2 [*ACS Appl. Mater. Interfaces* **10**, 42524 (2018), *J. Phys. D: Appl. Phys.* **55**, 345101 (2022)], generating defect could cause various unexpected phenomena such as defect-induced exciton state and the degradation of device efficiency.

Alternatively, we use the plasmon-induced hot electron injection, which increases the electron density near the nanogap. This acts as increasing defect density α without the afore mentioned unexpected side effects of generating defects. We have added the detailed description on the relation between α and n_e and the experimental way to control α in our system in the revised manuscript.

[Revised text] If we define this global defect density of the sample as α , then an increasing α indicates the provision of extra electrons, i.e., α is proportional to the electron density $n_e(x)$.

[Revised text] Subsequently, we gradually increase α and plot the evolution of the spatial distribution of the electrons $n_e(x)$ to investigate the electron density at the nanogap center when plasmon-induced hot electron generation occurs in the SPP mode.

[Added text] In our experiment, increasing α can be realized by plasmon-induced hot electrons, as it increases the background electron density near the nanogap.

7) The gap of the nanostructure (waveguide) is supposed to be SiO_2 . How to understand the work function of the waveguide compared to the gold in the center as shown in Fig. 5(a) and Fig. 5(b)?

Kelvin probe force microscopy measures contact potential difference (V_{CPD}) between the tip and sample. Given the relation $eV_{\text{CPD}} = \phi_{\text{tip}} - \phi_{\text{sample}}$, where ϕ_{tip} and ϕ_{sample} is the work function of tip and sample, ϕ_{sample} can be derived with already known ϕ_{tip} . Specifically, Kwon et al. [*Scientific Reports* **9**, 14434 (2019)] reported a decrease in work function of MoS₂ ML on the Au substrate due to the interfacial electric dipole energy. In this case, $\phi_{\text{sample}} = \phi_{\text{tip}} - (eV_{\text{CPD}} - \mu)$, where μ is the interfacial electric dipole energy, leading to the smaller work function compared to the suspended MoS₂, as shown in Fig. 5b-c. We acknowledge the lack of the description on the KPFM measurement. Consequently, we have added the detailed description of KPFM measurement in Methods section of the revised manuscript.

[Added text] We measured the work function of the region of a gap (suspended ML MoS₂) sample by using a conventional AFM (Park Systems Co., NX10) with a measurement mode of Kelvin probe force microscope (KPFM) addressing nanoscale surface work function. We used a commercial Cr & Au coated cantilever (Mikromasch Co., NSC18). Before the main measurement, we performed a calibration on the work function of cantilever by scanning the highly ordered pyrolytic graphite (HOPG, ~4.6 eV), commonly used as calibration method for KPFM. Then, we performed the nanoscale work function mapping on the suspended ML MoS₂ with the calibrated cantilever showing maximum value of about 5.4 eV. Note that the KPFM measurement was performed without optical excitation to precisely characterize the work function of MoS₂ ML without external perturbation.

8) For the description of the fabrication of nanogaps through focused ion-beam milling, it is not clear for me that why it needs two times deposit of Au. More detail statements are needed.

We apologize for the unclear description on the fabrication part. The coupling of light and SPP propagation rely on a MIM geometry where the top and sidewalls of the nanogap are gold-coated, whereas bottom of the channel is SiO₂. For fabrication of the channel, we start with a SOI wafer, and deposit 150 nm of Au on it. Then we perform a FIB milling step to etch into the SiO₂. As seen in the Fig. S3, after milling the resultant nanogaps have a thin layer of gold on top, but much of the sidewall is SiO₂. This milling process results in a slight taper to the sidewalls, which is crucial for subsequent gold deposition step. The top layer of gold is then removed using a gold etchant. A fresh layer of Au (50 nm) is deposited using e-beam, and coats the top, sidewalls, and bottom of the nanogap. A second round of milling is then performed to remove the gold from bottom of the nanogap channel. As the figure illustrates, the first milling process is an oxide etch step, whereas the second milling is an Au etch step to remove gold from bottom of the channels. We have added this detailed fabrication process in the Supplementary Information to deliver clear information to readers.

[Revised text] By contrast, the proposed nanogap-based lateral MIM waveguide device with the designed SPP mode can supply extra electrons locally via plasmon-induced hot electron generation, as illustrated in Fig. 1b (see Fig. S3 in Supplementary Information) [*ACS Photon.* **6**, 1500 (2019), *Nat. Photon.* **8**, 95 (2014)].

[Added results in SI]

Fig. S3. Fabrication process of lateral MIM waveguide.

The coupling of light and SPP propagation rely on a lateral MIM geometry where the top and sidewalls of the nanogap are gold-coated, whereas bottom of the channel is SiO₂. For fabrication of the channel, we start with a SOI wafer, and deposit 150 nm of Au on it. Then we perform a FIB milling step to etch into the SiO₂. As seen in the Fig. S3, after milling the resultant nanogaps have a thin layer of gold on top, but much of the sidewall is SiO₂. This milling process results in a slight taper to the sidewalls, which is crucial for subsequent gold deposition step. The top layer of gold is then removed using a gold etchant. A fresh layer of Au (50 nm) is deposited using e-beam, and coats the top, sidewalls, and bottom of the nanogap. A second round of milling is then performed to remove the gold from bottom of the nanogap channel. As the figure illustrates, the first milling process is an oxide etch step, whereas the second milling is an Au etch step to remove gold from bottom of the channels.

The parameters for fabricating lateral MIM waveguide, such as gap size and height, are optimized in the way of maximizing the efficiency of SPP coupling to the nanogap while also minimizing losses during SPP propagation in the nanogap. Specifically, the nanogap lateral MIM geometry (Au-SiO₂-Au) has been designed to provide a high contrast between the effective refractive index inside the channel and the refractive index of the substrate ($n_{\text{SiO}_2} = 1.46$) providing a higher coupling efficiency. Furthermore, coupling of incoming laser light with lateral MIM nanogap to activate SPP mode proceeds through scattering from the edges and sidewalls of the gap. This method where light is coupled to Au-SiO₂-Au MIM nanogap through edge/tail-end illumination has been discussed in detail in previous works [*Nature Photonics* **6**, 838 (2012), *Science* **338**, 1317 (2012), *Nature communications* **11**, 2930 (2020)].

9) For my understanding, the metal-insulator-metal structure in the work refers to the lateral structure. However, it is not clearly point out in the manuscript.

We thank the reviewer for pointing this out. We have thoroughly replaced the terminology “MIM waveguide structure” to “lateral MIM waveguide structure” in the revised manuscript.

Reviewers' Comments:

Reviewer #1:

Remarks to the Author:

I am satisfied with the author's response. They have well answered all the comments point-by-point. I recommend for accepting the manuscript.

Reviewer #2:

Remarks to the Author:

The authors have considered my comments and addressed my questions in their response letter. I am satisfied with authors' answers, and their corresponding revisions in response to my questions/suggestions. The manuscript has been improved for better presentation and understanding. I am pleased to recommend consideration of this manuscript to be published in Nature Communication.

Reviewer #3:

Remarks to the Author:

Lee et al. have conducted a study on the generation of high-purity trions and dynamic exciton-trion conversion using a lateral metal-insulator-metal (MIM) waveguide with spatial controllability of the surface plasmon polariton (SPP) mode facilitated with a spatial light modulator (SLM). They utilized a monolayer MoS₂ suspended on the nanogap geometry of the lateral plasmonic MIM waveguide, inducing a strain gradient that significantly increases the funneling efficiency for confining excitons to the nanogap. The authors also employed a SLM with adaptive wavefront shaping and sequence feedback algorithm to spatially control the trion emission intensity. The physical mechanism of electron funneling and the related exciton-to-trion conversion dynamics was analyzed by a drift-diffusion model and experimentally obtained Kelvin probe force microscopy (KPFM) data. The study provides insight into the potential for all-optical control of high-purity trions in nanoscale waveguides. The authors have done a good job of incorporating my suggestions and providing additional explanations to enhance the completeness and readability of the article. The article is well-suited for publication in Nature Communications, where it can serve as a valuable reference for researchers in the field.

We indicate the reviewers' comment in black, with our replies in blue.

Reviewer #1:

I am satisfied with the author's response. They have well answered all the comments point-by-point. I recommend for accepting the manuscript.

We thank the reviewer for the helpful comments to improve our manuscript. We are glad that the reviewer satisfactorily reads our answer and finds our manuscript suitable for publication.

Reviewer #2:

The authors have considered my comments and addressed my questions in their response letter. I am satisfied with authors' answers, and their corresponding revisions in response to my questions/suggestions. The manuscript has been improved for better presentation and understanding. I am pleased to recommend consideration of this manuscript to be published in Nature Communication.

We thank the reviewer for the helpful comments to improve our manuscript. We are glad that the reviewer finds improved qualities of our manuscript and recommends the publications.

Reviewer #3:

Lee et al. have conducted a study on the generation of high-purity trions and dynamic exciton-trion conversion using a lateral metal-insulator-metal (MIM) waveguide with spatial controllability of the surface plasmon polariton (SPP) mode facilitated with a spatial light modulator (SLM). They utilized a monolayer MoS₂ suspended on the nanogap geometry of the lateral plasmonic MIM waveguide, inducing a strain gradient that significantly increases the funneling efficiency for confining excitons to the nanogap. The authors also employed a SLM with adaptive wavefront shaping and sequence feedback algorithm to spatially control the trion emission intensity. The physical mechanism of electron funneling and the related exciton-to-trion conversion dynamics was analyzed by a drift-diffusion model and experimentally obtained Kelvin probe force microscopy (KPFM) data. The study provides insight into the potential for all-optical control of high-purity trions in nanoscale waveguides. The authors have done a good job of incorporating my suggestions and providing additional explanations to enhance the completeness and readability of the article. The article is well-suited for publication in Nature Communications, where it can serve as a valuable reference for researchers in the field.

We thank the reviewer for the help in improving the manuscript and the recommendation that our manuscript is suitable for publication.